# Dynamic 3D Gaussian Fields for Urban Areas

**Tobias Fischer**[1]  **Jonas Kulhanek**[1,3]  **Samuel Rota Bulò**[2]  **Lorenzo Porzi**[2]
**Marc Pollefeys**[1]  **Peter Kontschieder**[2]
[1] ETH Zürich   [2] Meta Reality Labs   [3] CTU Prague
https://tobiasfshr.github.io/pub/4dgf/

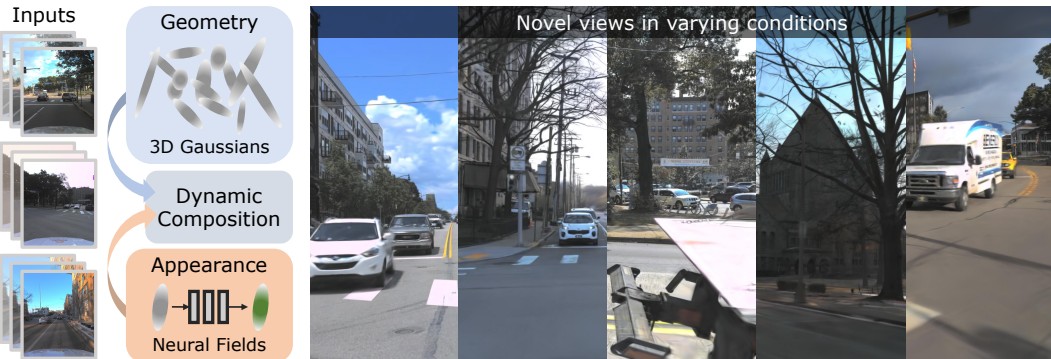

Figure 1: **Summary.** Given a set of *heterogeneous* input sequences that capture a common geographic area in varying environmental conditions (*e.g.* weather, season, and lighting) with distinct dynamic objects (*e.g.* vehicles, pedestrians, and cyclists), we optimize a *single* dynamic scene representation that permits rendering of arbitrary viewpoints and scene configurations at interactive speeds.

## Abstract

We present an efficient neural 3D scene representation for novel-view synthesis (NVS) in large-scale, dynamic urban areas. Existing works are not well suited for applications like mixed-reality or closed-loop simulation due to their limited visual quality and non-interactive rendering speeds. Recently, rasterization-based approaches have achieved high-quality NVS at impressive speeds. However, these methods are limited to small-scale, *homogeneous* data, *i.e.* they cannot handle severe appearance and geometry variations due to weather, season, and lighting and do not scale to larger, dynamic areas with thousands of images. We propose 4DGF, a neural scene representation that scales to large-scale *dynamic* urban areas, handles *heterogeneous* input data, and substantially improves rendering speeds. We use 3D Gaussians as an efficient geometry scaffold while relying on neural fields as a compact and flexible appearance model. We integrate scene dynamics via a scene graph at global scale while modeling articulated motions on a local level via deformations. This decomposed approach enables flexible scene composition suitable for real-world applications. In experiments, we surpass the state-of-the-art by over 3 dB in PSNR and more than $200\times$ in rendering speed.

## 1   Introduction

The problem of synthesizing novel views from a set of images has received widespread attention in recent years due to its importance for technologies like AR/VR and robotics. In particular, obtaining interactive, high-quality renderings of large-scale, dynamic urban areas under varying weather, lighting, and seasonal conditions is a key requirement for closed-loop robotic simulation and immersive VR experiences. To achieve this goal, sensor-equipped vehicles act as a frequent data

source that is becoming widely available in city-scale mapping and autonomous driving, creating the possibility of building up-to-date digital twins of entire cities. However, modeling these scenarios is extremely challenging as heterogeneous data sources have to be processed and combined: different weather, lighting, seasons, and distinct dynamic and transient objects pose significant challenges to the reconstruction and rendering of dynamic urban areas.

In recent years, neural radiance fields have shown great promise in achieving realistic novel view synthesis of static [1, 2, 3] and dynamic scenes [4, 5, 6, 7]. While earlier methods were limited to controlled environments, several recent works have explored large-scale, dynamic areas [8, 9, 10]. Among these, many works resort to removing dynamic regions and thus produce partial reconstructions [9, 10, 11, 12, 13, 14]. In contrast, fewer works model scene dynamics [15, 16, 17]. These methods exhibit clear limitations, such as rendering speed which can be attributed to the high cost of ray traversal in volumetric rendering.

Therefore, rasterization-based techniques [18, 19, 20, 11] have recently emerged as a viable alternative. Most notably, Kerbl *et al*. [18] propose a scene representation based on 3D Gaussian primitives that can be efficiently rendered with a tile-based rasterizer at a high visual quality. While demonstrating impressive rendering speeds, it requires millions of Gaussian primitives with high-dimensional spherical harmonics coefficients as color representation to achieve good view synthesis results. This limits its applicability to large-scale urban areas due to high memory requirements. Furthermore, due to its explicit color representation, it cannot model transient geometry and appearance variations commonly encountered in city-scale mapping and autonomous driving use cases such as seasonal and weather changes. Lastly, the approach is limited to static scenes which complicates representing dynamic objects such as moving vehicles or pedestrians commonly encountered in urban areas.

To this end, we propose 4DGF, a method that takes a hybrid approach to modeling dynamic urban areas. In particular, we use 3D Gaussian primitives as an efficient geometry scaffold. However, we do not store appearance as a per-primitive attribute, thus avoiding more than 80% of its memory footprint. Instead, we use fixed-size neural fields as a compact and flexible alternative. This allows us to model drastically different appearances and transient geometry which is essential to reconstructing urban areas from heterogeneous data. Finally, we model scene dynamics with a graph-based representation that maps dynamic objects to canonical space for reconstruction. We model non-rigid deformations in this canonical space with our neural fields to cope with articulated dynamic objects common in urban areas such as pedestrians and cyclists. This decomposed approach further enables a flexible scene composition suitable to downstream applications. The key contributions of this work are:

- We introduce 4DGF, a hybrid neural scene representation for dynamic urban areas that leverages 3D Gaussians as an efficient geometry scaffold and neural fields as a compact and flexible appearance representation.

- We use neural fields to incorporate scene-specific transient geometry and appearances into the rendering process of 3D Gaussian splatting, overcoming its limitation to static, homogeneous data sources while benefitting from its efficient rendering.

- We integrate scene dynamics via i) a graph-based representation, mapping dynamic objects to canonical space, and ii) modeling non-rigid deformations in this canonical space. This enables effective reconstruction of dynamic objects from in-the-wild captures.

We show that 4DGF effectively reconstructs large-scale, dynamic urban areas with over ten thousand images, achieves state-of-the-art results across four dynamic outdoor benchmarks [21, 22, 17, 23], and is more than $200\times$ faster to render than the previous state-of-the-art.

## 2   Related Work

**Dynamic scene representations.** Scene representations are a pillar of computer vision and graphics research [24]. Over decades, researchers have studied various static and dynamic scene representations for numerous problem setups [25, 26, 27, 28, 29, 30, 31, 32, 1, 33, 34]. Recently, neural rendering [35] has given rise to a new class of scene representations for photo-realistic image synthesis. While earlier methods in this scope were limited to static scenes [2, 3, 36, 37, 38], dynamic scene representations have emerged quickly [4]. These scene representations can be broadly classified into implicit and explicit representations. Implicit representations [5, 6, 7, 39, 4, 40, 41, 16] encode the scene as a parametric function modeled as neural network, while explicit representations [42, 43, 44, 45, 44] use

a collection of low-level primitives. In both cases, scene dynamics are simulated as i) deformations of a canonical volume [5, 6, 42, 39, 41], ii) particle-level motion such as scene flow [7, 4, 40, 16, 46], or iii) rigid transformations of local geometric primitives [44]. On the contrary, traditional computer graphics literature uses scene graphs to compose entities into complex scenes [47]. Therefore, another area of research explores decomposing scenes into higher-level elements [31, 48, 32, 15, 49, 50, 17], where entities and their spatial relations are expressed as a directed graph. This concept was recently revisited for view synthesis [15, 17]. In this work, we take a hybrid approach that uses i) explicit geometric primitives for fast rendering, ii) implicit neural fields to model appearance and geometry variation, and iii) a scene graph to decompose individual dynamic and static components.

**Efficient rendering and 3D Gaussian splatting.** Aside from accuracy, the rendering speed of a scene representation is equally important. While rendering speed highly depends on the representation efficiency itself, it also varies with the form of rendering that is coupled with it to generate an image [51]. Traditionally, neural radiance fields [3] use implicit functions and volumetric rendering which produce accurate renderings but suffer from costly function evaluation and ray traversal. To remedy these issues, many techniques for caching and efficient sampling [52, 53, 54, 36, 55, 17] have been developed. However, these approaches often suffer from excessive GPU memory requirements [52] and are still limited in rendering speed [54, 55, 17]. Therefore, researchers have opted to exploit more efficient forms of rendering, baking neural scene representations into meshes for efficient rasterization [19, 20, 11]. This area of research has recently been disrupted by 3D Gaussian splatting [18], which i) represents the scene as a set of anisotropic 3D Gaussian primitives ii) uses an efficient tile-based, differentiable rasterizer, and iii) enables effective optimization by adaptive density control (ADC), which facilitates primitive growth and pruning. This led to a paradigm shift from baking neural scene representations to a more streamlined approach.

However, the method of Kerbl. *et al.* [18] exhibits clear limitations, which has sparked a very active field of research with many concurrent works [44, 56, 57, 58, 59, 60, 61, 62, 63, 64, 65]. For instance, several works tackle dynamic scenes by adapting approaches described in the paragraph above [44, 66, 67, 68, 69, 70]. Another line of work focuses on modeling larger-scale scenes [65, 71, 72]. Lastly, several concurrent works investigate the reconstruction of dynamic street scenes [73, 74, 75]. These methods are generally limited to homogeneous data and in scale. In contrast, our method scales to tens of thousands of images and effectively reconstructs large, *dynamic* urban areas from *heterogeneous* data while *also* providing orders of magnitude faster rendering than traditional approaches.

**Reconstructing urban areas.** Dynamic urban areas are particularly challenging to reconstruct due to the complexity of both the scenes and the capturing process. Hence, significant research efforts have focused on adapting view synthesis approaches from controlled, small-scale environments to larger, real-world scenes. In particular, researchers have investigated the use of depth priors from *e.g.* LiDAR, providing additional information such as camera exposure, jointly optimizing camera parameters, and developing specialized sky and light modeling approaches [8, 9, 10, 11, 12, 13, 14]. However, since scene dynamics are challenging to approach, many works simply remove dynamic areas, providing only a partial reconstruction. A few works explicitly model scene dynamics, but suffer from limitations in terms of scalability [15, 49, 45], accuracy [16], rendering speed [17, 76, 77], or modeling of non-rigid and uncommon objects [15, 49, 45, 77, 17]. We introduce a mechanism to handle transient geometry and varying appearance, improve rendering efficiency, and, inspired by how global rigid object motion is handled in [17, 15, 49, 45, 77], propose an approach to model the local articulated motion of non-rigid dynamic objects without using semantic priors. Consequently, our work enables the reconstruction of much larger urban areas with a significantly higher number and diversity of dynamic objects across multiple in-the-wild captures.

## 3 Method

### 3.1 Problem setup

We are given a set of *heterogeneous* sequences $S$ that capture a common geographic area from a moving vehicle. The vehicle is equipped with calibrated cameras mounted in a surround-view setup. We denote with $C_s$ the set of cameras of sequence $s \in S$ and with $C$ the total set of cameras, *i.e.* $C := \bigcup_{s \in S} C_s$. For each camera $c \in C$, we assume to know the intrinsic $\mathsf{K}_c$ parameters and the pose $\mathsf{P}_c \in \mathrm{SE}(3)$, expressed in the ego-vehicle reference frame. Ego-vehicle poses $\mathsf{P}_s^t \in \mathrm{SE}(3)$ are provided for each sequence $s \in S$ and timesteps $t \in T_s$ and are expressed in the world reference

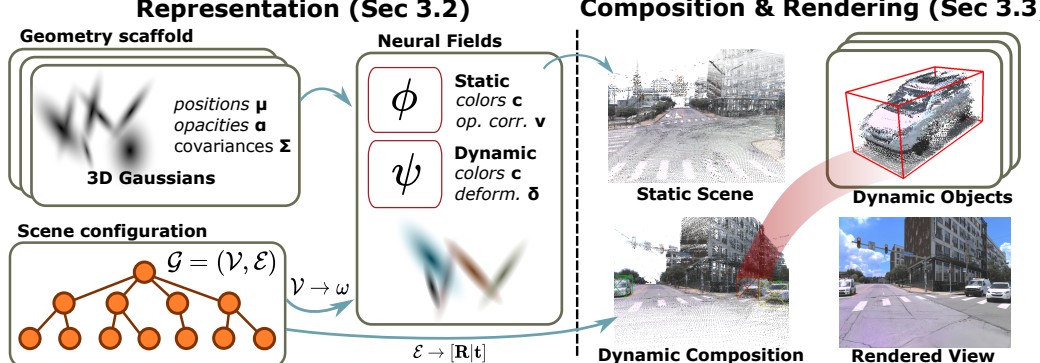

**Figure 2: Overview.** To render an image of sequence $s$ at time $t$, we first evaluate the scene graph $\mathcal{G} = (\mathcal{V}, \mathcal{E})$ which stores latent codes $\omega$ at its nodes $\mathcal{V}$ and coordinate transformations $[\mathbf{R}|\mathbf{t}]$ at its edges $\mathcal{E}$, *i.e.* the configuration of the dynamic objects and the overall scene. We then use the scene configuration to determine the active sets of 3D Gaussians $G$. The 3D Gaussians $G$ and the latent codes $\omega$ serve as conditioning signals to the neural fields $\phi$ and $\psi$, which output, for each 3D Gaussian $\mathfrak{g}_k \in G$, an appearance conditioned color $\mathbf{c}_k^{s,t}$, an opacity correction term $\nu_k^{s,t}$ for static Gaussians modeling transient geometry, and a dynamic deformation $\delta_k^t$ for non-rigid dynamic 3D Gaussians modeling *e.g.* pedestrians. Finally, the retrieved information is used to compose a set of 3D Gaussians that represent the dynamic scene at $(s,t)$ from which we render the image.

frame that is shared across all sequences. Here, $T_s$ denotes a set of timestamps relative to $s$. Indeed, we assume that timestamps cannot be compared across sequences because we lack a mapping to a global timeline, which is often the case with benchmark datasets due to privacy reasons. For each sequence $s \in S$, camera $c \in C_s$ and timestamp $t \in T_s$ we have an RGB image $\mathtt{I}_{(s,c)}^t \in [0,1]^{H \times W \times 3}$. Each sequence has additionally an associated set of dynamic objects $O_s$. Dynamic objects $o \in O_s$ are associated with a 3D bounding box track that holds its (stationary) 3D object dimensions $\mathbf{s}_o \in \mathbb{R}_+^3$ and poses $\{\xi_o^{t_0}, ..., \xi_o^{t_n}\} \subset \mathrm{SE}(3)$ w.r.t. the ego-vehicle frame, where $t_i \in T_o \subset T_s$. Our goal is to estimate the plenoptic function for the shared geographic area spanned by the training sequences, *i.e.* a function $f(\mathtt{P}, \mathtt{K}, t, s)$, which outputs a rendered RGB image of size $(H, W)$ for a given camera pose $\mathtt{P}$ with calibration $\mathtt{K}$ in the conditions of sequence $s \in S$ at time $t \in T_s$.

### 3.2 Representation

We model a parameterized, plenoptic function $f_\theta$, which depends on the following components: i) a scene graph $\mathcal{G}$ that provides the scene configuration and latent conditioning signals $\omega$ for each sequence $s$, object $o$, and time $t$, ii) sets of 3D Gaussians that serve as a geometry scaffold for the scene and objects, and iii) implicit neural fields that model appearance and modulate the geometry scaffold according to the conditioning signals. See Figure 2 for an overview of our method.

**Scene configuration.** Inspired by [17], we factorize the scene with a graph representation $\mathcal{G} = (\mathcal{V}, \mathcal{E})$, holding latent conditioning signals at the nodes $\mathcal{V}$ and coordinate system transformations along the edges $\mathcal{E}$. The nodes $\mathcal{V}$ consist of a root node $v_r$ defining the global coordinate system, *camera* nodes $\{v_c\}_{c \in C}$, and for each sequence $s \in S$, *sequence* nodes $\{v_s^t\}_{t \in T_s}$ and dynamic *object* nodes $\{v_o\}_{o \in O_s}$. We associate latent vectors $\omega$ to sequence and object nodes representing local appearance and geometry. Specifically, we model the time-varying sequence appearance and geometry via

$$\omega_s^t := [\mathtt{A}_s \gamma(t), \mathtt{G}_s \gamma(t)] \tag{1}$$

where $\mathtt{A}_s$ and $\mathtt{G}_s$ are appearance and geometry modulation matrices, respectively, and $\gamma(\cdot)$ is a 1D basis function of sines and cosines with linearly increasing frequencies at log-scale [35]. Time $t$ is normalized to $[-1, 1]$ via the maximum sequence length $\max_{s \in S} |T_s|$. For objects, we use both an object code and a time encoding

$$\omega_o^t := [\omega_o, \gamma(t)]. \tag{2}$$

Nodes in the graph $\mathcal{G}$ are connected by oriented edges that define rigid transformations between the canonical frames of the nodes. We have $\mathtt{P}_s^t$ for sequence to root edges, $\mathtt{P}_c$ for camera to sequence edges, and $\xi_o^t$ for object to sequence edges.

**3D Gaussians.** We represent the scene geometry with sets of anisotropic 3D Gaussians primitives $G = \{G_r\} \cup \{G_o : o \in O_s, s \in S\}$. Each 3D Gaussian primitive $\mathfrak{g}_k$ is parameterized by its mean

$\boldsymbol{\mu}_k$, covariance matrix $\Sigma_k$, and a base opacity $\alpha_k$. The covariance matrix is decomposed into a rotation matrix represented as a unit quaternion $q_k$ and a scaling vector $a_k \in \mathbb{R}^3_+$. The geometry of $\mathfrak{g}_k$ is represented by

$$\mathfrak{g}_k(\mathbf{x}) = \exp\left(-\frac{1}{2}[\mathbf{x} - \boldsymbol{\mu}_k]^\top \Sigma_k^{-1}[\mathbf{x} - \boldsymbol{\mu}_k]\right). \tag{3}$$

The common scene geometry scaffold is modeled with a single set of 3D Gaussians $G_r$, while we have a separate set $G_o$ of 3D Gaussians for each dynamic object $o$. Indeed, scene geometry is largely consistent across sequences while object geometries are distinct. The 3D Gaussians $G_r$ are represented in world frame, while each set $G_o$ is represented in a canonical, object-centric coordinate frame, which can be mapped to the world frame by traversing $\mathcal{G}$.

Differently from [18], our 3D Gaussians do not hold any appearance information, reducing the memory footprint of the representation *by more than 80%*. Instead, we leverage neural fields to regress a color information $\mathbf{c}_k^{s,t}$ and an updated opacity $\alpha_k^{s,t}$ for each sequence $s \in S$ and time $t \in T_s$. For 3D Gaussians in $G_r$ modeling the scene scaffolding, we predict an opacity attenuation term $\nu_k^{s,t}$ that is used to model transient geometry by downscaling $\alpha_k$. Instead, for 3D Gaussians in $G_o$ modeling objects the base opacity is left invariant. Hence

$$\alpha_k^{s,t} := \begin{cases} \nu_k^{s,t}\alpha_k & \text{if } \mathfrak{g}_k \in G_r \\ \alpha_k & \text{else.} \end{cases} \tag{4}$$

The attenuation term enforces a high base opacity for every 3D Gaussian visible in *at least* one sequence. Therefore, we can obtain pruning decisions in ADC by thresholding the base opacity $\alpha_k$, which is directly accessible without computational overhead, without risking the removal of transient geometry.

Furthermore, in the presence of non-rigid objects $o$, we predict deformation terms $\delta_k^t \in \mathbb{R}^3$ to the position of 3D primitives in $G_o$ via the neural fields, for each time $t \in T_o$. In this case, the position of the primitive in object-centric space at time $t$ is given by

$$\boldsymbol{\mu}_k^t := \boldsymbol{\mu}_k + \delta_k^t. \tag{5}$$

**Appearance and transient geometry.** Given the scene graph $\mathcal{G}$ and the 3D Gaussians $G$, we use two neural fields to decode the aforementioned parameters for each primitive. In particular, for 3D Gaussians in $G_r$ modeling the static scene, the neural field is denoted by $\phi$ and regresses the opacity attenuation term $\nu_k^{s,t}$ and a color $\mathbf{c}_k^{s,t}$, given the 3D Gaussian primitive's position $\boldsymbol{\mu}_k$, a viewing direction $\mathbf{d}$, the base opacity $\alpha_k$ and the latent code of the node $\omega_s^t$, *i.e.*

$$(\nu_k^{s,t}, \mathbf{c}_k^{s,t}) := \phi(\boldsymbol{\mu}_k, \mathbf{d}, \alpha_k, \omega_s^t). \tag{6}$$

where $s \in S$ and $t \in T_s$. Note that since the opacity attenuation $\nu_k^{s,t}$ contributes to modeling transient geometry by removing parts of the scene encoded in the original set of Gaussians, it does not depend on the viewing direction $\mathbf{d}$.

For 3D Gaussians in $G_o$ modeling dynamic objects, the neural field is denoted by $\psi$ and regresses a color $\mathbf{c}_k^{s,t}$. Besides the primitive's position and viewing direction, we condition $\psi$ on latent vectors $\omega_s^t$ and $\omega_o^t$ to model both local object texture and global sequence appearance such as illumination. Here, the sequence $s$ is the one where $o$ belongs to, *i.e.* satisfying $o \in O_s$, and $t \in T_o$. Accordingly, the color $\mathbf{c}_k^{s,t}$ for a 3D Gaussian in $G_o$ is given by

$$\mathbf{c}_k^{s,t} := \psi(\boldsymbol{\mu}_k, \mathbf{d}, \omega_s^t, \omega_o^t). \tag{7}$$

Both $\boldsymbol{\mu}_k$ and $\mathbf{d}$ are expressed in the canonical, object-centric space of object $o$. Using neural fields has three key advantages for our purpose. First, by sharing the parameters of $\phi$ and $\psi$ across all 3D Gaussians $G$, we achieve a significantly more compact representation than in [18] when scaling to large-scale urban areas. Second, it allows us to model sequence-dependent appearance and transient geometry which is fundamental to learning a scene representation from heterogeneous data. Third, information sharing between nodes enables an interaction of sequence and object appearance.

However, querying a neural field is more complex than a spherical harmonics function as in [18]. Therefore, we i) use efficient hash-grid representations [53] to minimize query complexity and, ii)

carefully optimize the rendering workflow to minimize the amount of queries. In particular, we skip out-of-view 3D Gaussians and implement a vectorized query function to $\psi$ that retrieves the parameters of all relevant dynamic objects in parallel. We refer to Section 4.3 for a runtime analysis.

**Non-rigid objects.** Street scenes are occupied not only by rigidly moving vehicles but also by, *e.g.*, pedestrians and cyclists that move in a non-rigid manner. These pose a significant challenge due to their unconstrained motion under limited visual coverage. Therefore, we take a decomposed approach to modeling non-rigid objects. First, we represent the local articulated motion of non-rigid objects like pedestrians as deformation in canonical space. We use deformation head $\chi$ that predicts a local position offset $\delta_k^t$ via

$$\delta_k^t := \chi(\mathbf{f}_\psi, \gamma(t)) \tag{8}$$

given an intermediate feature representation $\mathbf{f}_\psi$ of $\psi$ conditioned on $\boldsymbol{\mu}_k$ and time $t$. We deform the position of $\boldsymbol{\mu}_k$ over time in canonical space as per Equation (5). Second, we use the scene graph $\mathcal{G}$ to model the global rigid object motion, transforming the objects from object-centric to world space with a rigid body transformation. We use a general design to cover a wide range of scenarios, such as pedestrians holding a stroller or shopping bags, cyclists, and animals. See Figure 10 in our supp. mat.

**Background modeling.** To achieve a faithful rendering of far-away objects and the sky, it is important to have a background model. Inspired by [54], where points are sampled along a ray at increasing distance outside the scene bounds, we place 3D Gaussians on spheres around the scene with radius $r2^{i+1}$ for $i \in \{1, 2, 3\}$ where $r$ is half of the scene bound diameter. To avoid ambiguity with foreground scene geometry and to increase efficiency, we remove all points that are i) below the ground plane, ii) occluded by foreground scene points, or iii) outside of the view frustum of any training view. To uniformly distribute points on each sphere, we utilize the Fibonacci sphere sampling algorithm [78], which arranges points in a spiral pattern using a golden ratio-based formula. Even though this sampling is not optimal, it serves as a faster approximation of the optimal sampling.

## 3.3 Composition and Rendering

**Scene composition.** To render our representation from the perspective of camera $c$ at time $t$ in sequence $s$, we traverse the graph $\mathcal{G}$ to obtain the latent vector $\omega_s^t$ and the latent vector $\omega_o^t$ of each visible object $o \in O_s$, *i.e.* such that $t \in T_o$. Moreover, for each 3D Gaussian primitive $\mathfrak{g}_k$ in $G$, we use the collected camera parameters, object scale, and pose information to determine the transformation $\Pi_k^c$ mapping points from the primitive's reference frame (*e.g.* world for $G_r$, object-space for $G_o$) to the image space of camera $c$. Opacities $\alpha_k^{s,t}$ are computed as per Equation (4), while colors $\mathbf{c}_k^{s,t}$ are computed for primitives in $G_r$ and in $G_o$ via Equations (6) and (7), respectively. For non-rigid objects in $G_o$, we compute the primitive positions $\boldsymbol{\mu}_k^t$ via Equation (5).

**Rasterization.** To render the scene from camera $c$, we follow [18] and splat the 3D Gaussians to the image plane. Practically, for each primitive, we compute a 2D Gaussian kernel denoted by $\mathfrak{g}_k^c$ with mean $\boldsymbol{\mu}_k^c$ given by the projection of the primitive's position to the image plane, *i.e.* $\boldsymbol{\mu}_k^c := \Pi_k^c(\boldsymbol{\mu}_k)$, and with covariance given by $\Sigma_k^c := \mathsf{J}_k^c \Sigma_k \mathsf{J}_k^{c\top}$, where $\mathsf{J}_k^c$ is the Jacobian of $\Pi_k^c$ evaluated at $\boldsymbol{\mu}_k$. Finally, we apply traditional alpha compositing of the 3D Gaussians to render pixels $\mathbf{p}$ of camera $c$:

$$\mathbf{c}^{s,t}(\mathbf{p}) := \sum_{k=0}^{K} \mathbf{c}_k^{s,t} w_k \prod_{j=0}^{k-1}(1 - w_j) \qquad \text{with} \quad w_k := \alpha_k^{s,t} \mathfrak{g}_k^c(\mathbf{p}). \tag{9}$$

## 3.4 Optimization

To optimize parameters $\theta$ of $f_\theta$, *i.e.* 3D Gaussian parameters $\boldsymbol{\mu}_k$, $\alpha_k$, $q_k$ and $a_k$, sequence latent vectors $\omega_s^t$ and implicit neural fields $\psi$ and $\phi$, we use an end-to-end differentiable rendering pipeline. We render both an RGB color image $\hat{\mathtt{I}}$ and a depth image $\hat{\mathcal{D}}$ and apply the following loss function:

$$\mathcal{L}(\hat{\mathtt{I}}, \mathtt{I}, \hat{\mathcal{D}}, \mathcal{D}) = \lambda_{\text{rgb}}\mathcal{L}_{\text{rgb}}(\hat{\mathtt{I}}, \mathtt{I}) + \lambda_{\text{ssim}}\mathcal{L}_{\text{ssim}}(\hat{\mathtt{I}}, \mathtt{I}) + \lambda_{\text{dep}}\mathcal{L}_{\text{dep}}(\hat{\mathcal{D}}, \mathcal{D}) \tag{10}$$

where $\mathcal{L}_{\text{rgb}}$ is the L1 norm, $\mathcal{L}_{\text{ssim}}$ is the structural similarity index measure [79], and $\mathcal{L}_{\text{dep}}$ is the L2 norm. We use the posed training images and LiDAR measurements as the ground truth. If no depth ground-truth is available, we drop the depth-related loss from $\mathcal{L}$.

**Pose optimization.** Next to optimizing scene geometry, it is crucial to refine the pose parameters of the reconstruction for in-the-wild scenarios since provided poses often have limited accuracy [10, 17].

| Method | Residential | | | Downtown | | | Mean | | | Render (s) |
| | PSNR ↑ | SSIM ↑ | LPIPS ↓ | PSNR ↑ | SSIM ↑ | LPIPS ↓ | PSNR ↑ | SSIM ↑ | LPIPS ↓ | |
|---|---|---|---|---|---|---|---|---|---|---|
| Nerfacto + Emb. | 19.83 | 0.637 | 0.562 | 18.05 | 0.655 | 0.625 | 18.94 | 0.646 | 0.594 | 11.5 |
| Nerfacto + Emb. + Time | 20.05 | 0.641 | 0.562 | 18.66 | 0.656 | 0.603 | 19.36 | 0.654 | 0.583 | 11.6 |
| SUDS [16] | 21.76 | 0.659 | 0.556 | 19.91 | 0.665 | 0.645 | 20.84 | 0.662 | 0.601 | 74.0 |
| ML-NSG [17] | 22.29 | 0.678 | 0.523 | 20.01 | 0.681 | 0.586 | 21.15 | 0.680 | 0.555 | 21.7 |
| **4DGF (Ours)** | **25.78** | **0.772** | **0.405** | **24.16** | **0.772** | **0.488** | **24.97** | **0.772** | **0.447** | 0.074 |

Table 1: **Novel view synthesis on Argoverse 2 [81].** Our method improves substantially over the state-of-the-art while being more than $200\times$ faster to render at the original $1550 \times 2048$ resolution.

Thus, we optimize the residuals $\Delta P_s^t \in \mathfrak{se}(3)$, $\Delta P_c \in \mathfrak{se}(3)$ and $\Delta \xi_o^t \in \mathfrak{se}(2)$ jointly with parameters $\theta$. We constrain object pose residuals to $\mathfrak{se}(2)$ to incorporate the prior that objects move on the ground plane and are oriented upright. See our supp. mat. for details on camera pose gradients.

**Adaptive density control.** To facilitate the growth and pruning of 3D Gaussian primitives, the optimization of the parameters $\theta$ is interleaved by an ADC mechanism [18]. This mechanism is essential to achieve photo-realistic rendering. However, it was not designed for training on tens of thousands of images, and thus we develop a streamlined multi-GPU version of it. We accumulate statistics across processes and, instead of running ADC on GPU 0 and synchronizing the results, we synchronize only non-deterministic parts of ADC, *i.e.* the random samples drawn from the 3D Gaussians that are being split. These are usually much fewer than the total number of 3D Gaussians and thus avoids communication overhead. Next, the 3D Gaussian parameters are replaced by their updated replicas. However, this will impair the synchronization of the gradients because, in PyTorch DDP [80], parameters are only registered once at model initialization. Therefore, we re-initialize the `Reducer` upon finishing the ADC mechanism in the low-level API provided in [80].

Furthermore, urban street scenes pose some unique challenges to ADC, such as a large variation in scale, *e.g.* extreme close-ups of nearby cars mixed with far-away buildings and sky. This can lead to blurry renderings for close-ups due to insufficient densification. We address this by using maximum 2D screen size as a splitting criterion.[1] In addition, ADC considers the world-space scale $a_k$ of a 3D Gaussian to prune large primitives which hurts background regions far from the camera. Hence, we first test if a 3D Gaussian is inside the scene bounds before pruning it according to $a_k$. Finally, the scale of urban areas leads to memory issues when the growth of 3D Gaussian primitives is unconstrained. Therefore, we introduce a threshold that limits primitive growth while keeping pruning in place. See our supp. mat. for more details and analysis.

# 4 Experiments

**Datasets and metrics.** We evaluate our approach across various dynamic outdoor benchmarks. First, we utilize the recently proposed NVS benchmark [17] of Argoverse 2 [81] to compare against the state-of-the-art in the multi-sequence scenario and to showcase the scalability of our method. Second, we use the established Waymo Open [23], KITTI [21] and VKITTI2 [22] benchmarks to compare to existing approaches in single-sequence scenarios. For Waymo, we use the dynamic-32 split of [76], while for KITTI and VKITTI2 we follow [16, 17]. We apply commonly used metrics to measure view synthesis quality: PSNR, SSIM [79], and LPIPS (AlexNet) [82].

**Implementation details.** We use $\lambda_{rgb} := 0.8$, $\lambda_{ssim} := 0.2$ and $\lambda_{depth} := 0.05$. We use the LiDAR point clouds as initialization for the 3D Gaussians. We first filter the points of dynamic objects using the 3D bounding box annotations and subsequently initialize the static scene with the remaining points while using the filtered points to initialize each dynamic object. We use mean voxelization with voxel size $\tau$ to remove redundant points. See our supp. mat. for more details.

## 4.1 Comparison to state-of-the-art

We compare with prior art across two experimental settings: *single-sequence* and *multi-sequence*. In the former, we are given a single input sequence and aim to synthesize hold-out viewpoints from that same sequence. In the latter, we are given *multiple, heterogeneous* input sequences and aim to synthesize hold-out viewpoints across *all* of these sequences from a *single* model.

---

[1]Note that while this criterion was described in [18], it was not used in the experiments.

| Method | KITTI [75%] | | | KITTI [50%] | | | KITTI [25%] | | |
|---|---|---|---|---|---|---|---|---|---|
| | PSNR ↑ | SSIM ↑ | LPIPS ↓ | PSNR ↑ | SSIM ↑ | LPIPS ↓ | PSNR ↑ | SSIM ↑ | LPIPS ↓ |
| NSG [15] | 21.53 | 0.673 | 0.254 | 21.26 | 0.659 | 0.266 | 20.00 | 0.632 | 0.281 |
| SUDS [16] | 22.77 | 0.797 | 0.171 | 23.12 | 0.821 | 0.135 | 20.76 | 0.747 | 0.198 |
| MARS [83] | 24.23 | 0.845 | 0.160 | 24.00 | 0.801 | 0.164 | 23.23 | 0.756 | 0.177 |
| StreetGaussians [73] | 25.79 | 0.844 | 0.081 | 25.52 | 0.841 | 0.084 | 24.53 | 0.824 | 0.090 |
| ML-NSG [17] | 28.38 | 0.907 | 0.052 | 27.51 | 0.898 | 0.055 | 26.51 | 0.887 | 0.060 |
| **4DGF (Ours)** | **31.34** | **0.945** | **0.026** | **30.55** | **0.931** | **0.028** | **29.08** | **0.908** | **0.036** |

| Method | VKITTI2 [75%] | | | VKITTI2 [50%] | | | VKITTI2 [25%] | | |
|---|---|---|---|---|---|---|---|---|---|
| | PSNR ↑ | SSIM ↑ | LPIPS ↓ | PSNR ↑ | SSIM ↑ | LPIPS ↓ | PSNR ↑ | SSIM ↑ | LPIPS ↓ |
| NSG [15] | 23.41 | 0.689 | 0.317 | 23.23 | 0.679 | 0.325 | 21.29 | 0.666 | 0.317 |
| SUDS [16] | 23.87 | 0.846 | 0.150 | 23.78 | 0.851 | 0.142 | 22.18 | 0.829 | 0.160 |
| MARS [83] | 29.79 | 0.917 | 0.088 | 29.63 | 0.916 | 0.087 | 27.01 | 0.887 | 0.104 |
| StreetGaussians [73] | 30.10 | 0.935 | **0.025** | 29.91 | 0.932 | **0.026** | 28.52 | 0.917 | **0.034** |
| ML-NSG [17] | 29.73 | 0.912 | 0.065 | 29.19 | 0.906 | 0.066 | 28.29 | 0.901 | 0.067 |
| **4DGF (Ours)** | **30.67** | **0.943** | 0.035 | **30.45** | **0.939** | 0.036 | **29.27** | **0.923** | 0.041 |

Table 2: **Novel view synthesis on KITTI [21] and VKITTI2 [22].** Our method produces state-of-the-art results across both benchmarks and at varying training view fractions.

**Multi-sequence setting.** In Table 1, we show results on the Argoverse 2 NVS benchmark proposed in [17]. We compare to state-of-the-art approaches [16, 17] and the baselines introduced in [17]. The results highlight that our approach scales well to large-scale dynamic urban scenes, outperforming previous work in performance and rendering speed by a significant margin. Specifically, we outperform [17] by more than 3 dB in PSNR while rendering more than $200\times$ faster. To examine these results more closely, we show a qualitative comparison in Figure 3. We see that while SUDS [16] struggles with dynamic objects and ML-NSG [17] produces blurry renderings, our work provides sharp renderings and accurately represented dynamic objects, in both RGB color and depth images. Overall, the results highlight that our model can faithfully represent heterogeneous data at high visual quality in a single 3D representation while being much faster to render than previous work.

**Single-sequence setting.** In Table 2, we show results on the KITTI [21] and VKITTI [22] benchmarks at varying training view fractions, *i.e.* from dense towards sparse view setting. Furthermore, we follow the experimental protocol in [77] and show an additional comparison on the KITTI dataset with a different data split that uses also approx. 50% of views for training in Table 3. Our approach consistently outperforms previous work as well as concurrent 3D Gaussian-based approaches [73]. In Table 4, we show results on Waymo Open [23], specifically on the

| Method | PSNR ↑ | SSIM ↑ | LPIPS ↓ |
|---|---|---|---|
| SUDS† [16, 83] | 23.12 | 0.821 | 0.135 |
| MARS [83] | 24.00 | 0.801 | 0.164 |
| NeuRAD [77] | 27.00 | 0.795 | 0.082 |
| NeuRAD-2x [77] | 27.91 | 0.822 | 0.066 |
| **4DGF (Ours)** | **30.01** | **0.913** | **0.052** |

Table 3: **Novel view synthesis on KITTI [21] on split from [77].** We follow the experimental protocol in [77] to compare to additional works. †baseline from [83].

Dynamic-32 split proposed in [76]. We outperform previous work by a large margin while our rendering speed is $700\times$ faster than the best alternative. Note that our rendering speed increases for smaller-scale scenes. Furthermore, we show that, contrary to previous approaches, our method does not suffer from lower view quality in dynamic areas. This corroborates the strength of our contributions, showing that our method is not only scalable to large-scale, heterogeneous street data but also demonstrates superior performance in smaller-scale, homogeneous street data.

## 4.2 Ablation studies

We verify our design choices in both multi- and single-sequence settings. For a fair comparison, we set the global maximum of 3D Gaussians to 8 and 4.1 million, respectively. We perform these ablation studies on the residential split of [17]. We use the full overlap in the multi-sequence setting, while using a single sequence of this split for the single-sequence setting. In Table 5a, we verify

| Method | Full Image | | Dynamic-Only | | Render (s) |
|---|---|---|---|---|---|
| | PSNR ↑ | SSIM ↑ | PSNR ↑ | SSIM ↑ | |
| D²NeRF [41] | 24.17 | 0.642 | 21.44 | 0.494 | - |
| HyperNeRF [39] | 24.71 | 0.682 | 22.43 | 0.554 | - |
| EmerNeRF [76] | 27.62 | 0.792 | 24.18 | 0.670 | 18.9 |
| **4DGF (Ours)** | **29.04** | **0.881** | **29.34** | **0.884** | 0.025 |

Table 4: **Novel view synthesis on Waymo Open [23].** We use the Dynamic-32 split [76]. Contrary to prior work, we do not exhibit lower view quality in dynamic areas.

the components that are not specific to the multi-sequence setting. In particular, we show that our approach to modeling scene dynamics is highly effective, evident from the large disparity in performance between the static and the dynamic variants. Next, we show that modeling appearance with a neural field is on par with the proposed solution in [18], while

| Dynamic | Neural Fields | Background | PSNR ↑ | SSIM ↑ | LPIPS ↓ | GPU Mem. |
|---|---|---|---|---|---|---|
| - | - | - | 24.31 | 0.814 | 0.287 | 8.6 GB |
| ✓ | - | - | 28.55 | 0.839 | 0.262 | 8.6 GB |
| ✓ | ✓ | - | 28.49 | 0.838 | 0.262 | 4.5 GB |
| ✓ | ✓ | ✓ | **28.81** | **0.845** | **0.260** | 4.5 GB |

(a) Single-sequence.

| $\omega_s$ | $\alpha_s$ | PSNR ↑ | SSIM ↑ | LPIPS ↓ |
|---|---|---|---|---|
| - | - | 22.37 | 0.741 | 0.456 |
| ✓ | - | 25.13 | 0.767 | 0.412 |
| ✓ | ✓ | **25.78** | **0.772** | **0.405** |

(b) Multi-sequence.

Table 5: **Ablation studies.** We show that (a) our approaches to modeling scene dynamics and background regions are effective and neural fields are on-par with spherical harmonics while more memory efficient to train, and (b) using implicit fields for appearance *and* geometry is crucial for the multi-sequence setting. We control for the maximum number of 3D Gaussians for fair comparison.

| Dataset
Resolution
Avg. number of 3D Gaussians | Argoverse 2 [81]
$1550 \times 2048$
8.02M | Waymo Open [23]
$640 \times 960$
2.75M | Mean (ms) | Percentage (%) |
|---|---|---|---|---|
| 1. Scene graph evaluation: retrieve $\omega$, $[\mathbf{R}|\mathbf{t}]$, 3D Gaussians at $(s,t)$ | 38.5 | 13.0 | 25.75 | 52.4 |
| 2. Scene composition: apply $[\mathbf{R}|\mathbf{t}]$ to 3D Gaussians | 2.3 | 1.5 | 1.90 | 3.9 |
| 3. 3D Gaussian projection | 2.0 | 3.5 | 2.75 | 5.6 |
| 4. Query neural fields $\phi$ and $\psi$ | 9.5 | 3.6 | 6.55 | 13.3 |
| 5. Rasterization | 21.3 | 3.0 | 12.15 | 24.8 |
| Total | 73.6 | 24.6 | 49.1 | 100 |
| FPS | 13.6 | 40.7 | 20.4 | - |

Table 6: **Inference runtime analysis**. We report the individual and average timings of our method's components on two datasets. Overall, scene graph evaluation (1.) and rasterization (5.), dominate the runtime. While total runtime correlates with scene scale and image resolution, we achieve interactive frame rates on both datasets.

being more memory efficient. In particular, when modeling view-dependent color as a per-Gaussian attribute as in [18] the model uses 8.6 GB of peak GPU memory during training, while it uses only 4.5 GB with fixed-size neural fields. Similarly, storing the parameters of the former takes 922 MB, while the latter takes only 203 MB. Note that this disparity increases with the number of 3D Gaussians per scene. Finally, we achieve the best performance when adding the generated 3D Gaussian background.

We now scrutinize components specific to multi-sequence data in Table 5b. We compare the view synthesis performance of our model when i) not modeling sequence appearance or transient geometry, ii) only modeling sequence appearance, iii) modeling both sequence appearance *and* transient geometry. Naturally, we observe a large gap in performance between i) and ii), since the appearance changes between sequences are drastic (see Figure 3). However, there is still a significant gap between ii) and iii), demonstrating that modeling both sequence appearance *and* transient geometry is important for view synthesis from heterogeneous data sources. Finally, we provide qualitative results for non-rigid object view synthesis in Figure 4, and show that our approach can model articulate motion without the use of domain priors. In our supp. mat., we provide further analysis.

## 4.3 Runtime analysis

We divide our algorithm into its main components and report the individual inference runtimes across two datasets in Table 6. While the runtime clearly correlates with scene complexity and image resolution, we observe that, on average, the runtime is dominated by scene graph evaluation and rasterization, accounting for more than 75% of the total runtime. This owes to the complexity of rasterizing millions of primitives across a high-resolution image which is computationally demanding even for efficient rasterization algorithms [18], and handling hundreds to thousands of dynamic objects across one or multiple dynamic captures, making the retrieval of the 3D Gaussians and latent codes costly. In contrast, the queries to the neural fields account for only 13.3% of the average total runtime, making it a viable alternative to the spherical harmonics function in [18]. Overall, our method achieves interactive rendering speeds on both datasets and 20.4 FPS on average.

## 5 Conclusion

We presented 4DGF, a neural scene representation for dynamic urban areas. 4DGF models highly dynamic, large-scale urban areas with 3D Gaussians as efficient geometry scaffold and compact but flexible neural fields modeling large appearance and geometry variations across captures. We use a scene graph to model dynamic object motion and flexibly compose the representation at arbitrary

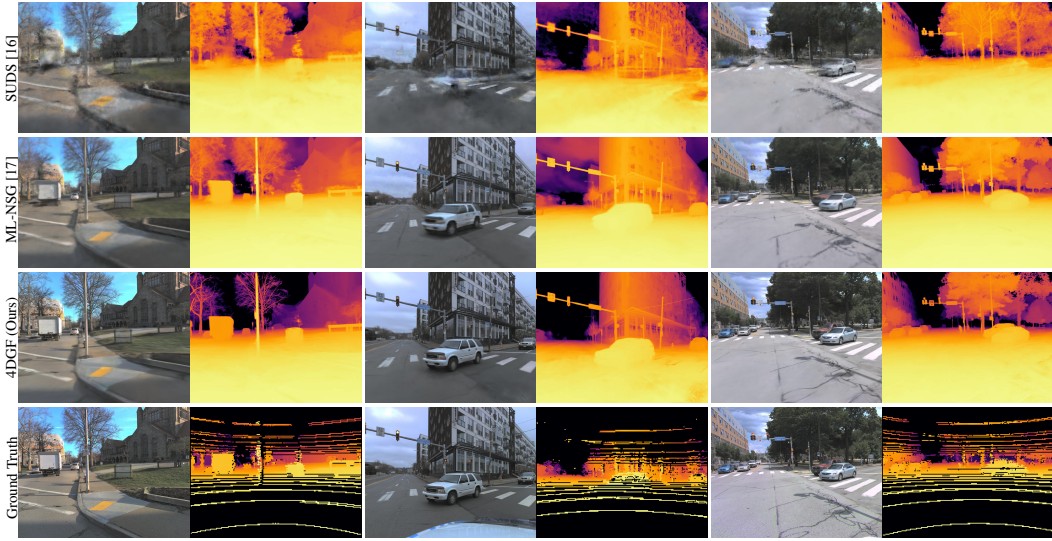

Figure 3: **Qualitative results on Argoverse 2 [81].** Our method produces significantly sharper renderings both in foreground dynamic and static background regions, with much fewer artifacts *e.g.* in areas with transient geometry such as tree branches (left). Best viewed digitally.

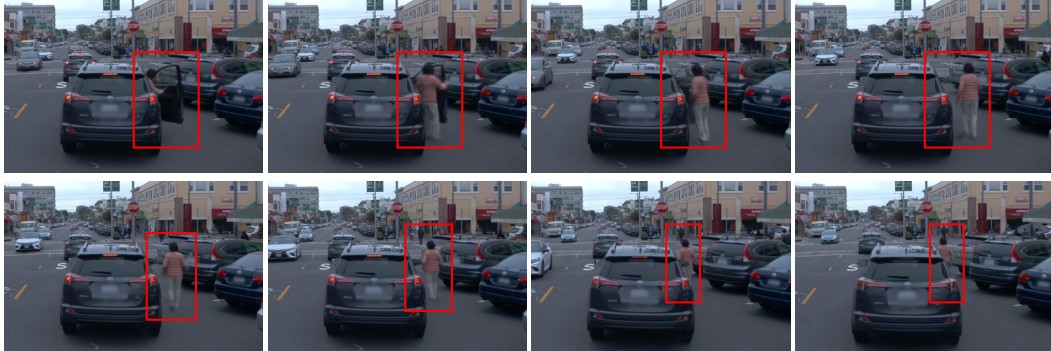

Figure 4: **Qualitative results on Waymo Open [23].** We show a sequence of evaluation views synthesized by our model (top-left to bottom-right). As the woman (marked with a red box) gets out of the car and walks away, we successfully model her articulated motion and changing body poses.

configurations and conditions. We jointly optimize the 3D Gaussians, the neural fields, and the scene graph, showing state-of-the-art view synthesis quality and interactive rendering speeds.

**Limitations and future work.** While 4DGF improves novel view synthesis in dynamic urban areas, the challenging nature of the problem leaves room for further exploration. Although we model scene dynamics, appearance, and geometry variations, other factors influence image renderings in real-world captures. First, in-the-wild captures often exhibit distortions caused by the physical image formation process. Therefore, modeling phenomena like rolling shutter, white balance, motion and defocus blur, or chromatic aberrations is necessary to avoid reconstruction artifacts. Second, the assumption of a pinhole camera model in [18] persists in our work. Thus, our method falls short of modeling more complex camera models like equirectangular cameras and other sensors such as LiDAR, which may be limiting for certain capturing or simulation settings.

**Broader impact.** We expect our work to positively impact real-world use cases like robotic simulation and mixed reality by improving the underlying technology. While we do not expect malicious uses of our method, we note that an inaccurate simulation, *i.e.* a failure of our system, could misrepresent the robotic system performance, possibly affecting real-world deployment.

# 6    Acknowledgements

The authors thank Haithem Turki, Songyou Peng, Erik Sandström, François Darmon, and Jonathon Luiten for useful discussions. Tobias Fischer was supported by a Meta SRA.

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

# A  Appendix

We provide further details on our method and the experimental setting, as well as additional experimental results. We accompany this supplemental material with a demonstration video.

## A.1  Demonstration Video

We showcase the robustness of our method by rendering a complete free-form trajectory across five highly diverse sequences *using the same model*. Specifically, we chose the model trained on the residential split in Argoverse 2 [81].

To obtain the trajectory, we interpolate keyframes selected throughout the total geographic area of the residential split into a single, smooth trajectory that encompasses most of its spatial extent. We also apply periodical translations and rotations to this trajectory to increase the variety of synthesized viewpoints. We use a constant speed of 10 meters per second. We choose five different sequences in the data split as the references, spanning sunny daylight conditions in summer to near sunset in winter. Consequently, the appearance of the sequences changes drastically, *e.g.* from green, fully-leafed trees to empty branches and snow or from bright sunlight to dark clouds. Furthermore, we render each sequence with its unique set of dynamic objects, simulating various distinct traffic scenarios.

We show that our model is able to perform dynamic view synthesis in all of these conditions at high quality, faithfully representing scene appearance, transient geometry, and dynamic objects in each of the conditions. We highlight that this scenario is *extremely* difficult, as it requires the model to generalize well beyond the training trajectories, represent totally different appearances and geometry, and model hundreds of dynamic, fast-moving objects. Despite this fact, our method produces realistic renderings, showing its potential for real-world applications.

## A.2  Method

**Neural field architectures.**  To maximize efficiency, we model $\phi$ and $\psi$ with hash grids and tiny MLPs [53]. The hash grids interpolate feature vectors at the nearest voxel vertices at multiple levels. The feature vectors are obtained by indexing a feature table with a hash function. Both neural fields are given input conditioning signals $\omega_s^t \in \mathbb{R}^{64}$ and $\omega_o^t := [\omega_o \in \mathbb{N}, \gamma(t) \in \mathbb{R}^{13}]$ and output a color $\mathbf{c}$ among the other outputs defined in Section 3.2.

For $\phi$, we use the 3D Gaussian mean $\boldsymbol{\mu}_k$ to query the hash function at a certain 3D position yielding an intermediate feature representation $\mathbf{f}_\phi$. We input the feature $\mathbf{f}_\phi$, the sequence latent code $\omega_s^t$, and the base opacity $\alpha_k$ into $\mathrm{MLP}_\alpha$ which outputs the opacity attenuation $\nu_k^{s,t}$. In a parallel branch, we input $\mathbf{f}_\phi$, $\omega_s^t$, and the viewing direction $\mathbf{d}$ encoded by a spherical harmonics encoding of degree 4 into the color head $\mathrm{MLP}_\mathbf{c}$ of $\phi$ that will define the final color of the 3D Gaussian.

For $\psi$, we use a 4D hash function while using only three dimensions for interpolation of the feature vectors, effectively modeling a 4D hash grid. We use both the position $\boldsymbol{\mu}_k$ and the object code $\omega_o$, *i.e.* the object identity, as the fourth dimension of the hash grid to model an arbitrarily large number of objects with a single hash table [16] *without a linear increase in memory*.

We input the intermediate feature $\mathbf{f}_\psi$ and the time encoding $\gamma(t)$ into the deformation head $\mathrm{MLP}_\chi$ which will output the non-rigid deformation of the object at time $t$, if applicable. In parallel, we input $\omega_s^t$, $\mathbf{f}_\psi$, $\gamma(t)$, and the encoded relative viewing direction $\mathbf{d}$ into the color head $\mathrm{MLP}_\mathbf{c}$ to output the final color. Note that relative viewing direction refers to the viewing direction in canonical, object-centric space. As noted in Section 3.2, the MLP heads are shared across all objects.

We list a detailed overview of the architectures in Table 7. Note that, i) we decrease the hash table size of $\psi$ in single-sequence experiments to $2^{15}$ as we find this to be sufficient, and ii) we use two identical networks for $\psi$ to separate rigid from non-rigid object instances.

**Color prediction.**  The kernel function $\mathfrak{g}_k$ prevents a full saturation of the rendered color within the support of the primitive as long as the primitive's RGB color is bounded in the $[0, 1]$ range. This can be a problem for background and other uniformly textured regions that contain large 3D Gaussians, specifically larger than a single pixel. Therefore, inspired by [84], we use a scaled sigmoid activation

| | Hash Table | | | MLP$_{\mathbf{c}}$ | | MLP$_\alpha$ | | MLP$_\chi$ | |
| | Size | # levels | max. res. | # layers | # neurons | # layers | # neurons | # layers | # neurons |
|---|---|---|---|---|---|---|---|---|---|
| $\phi$ | $2^{19}$ | 16 | 2048 | 3 | 64 | 2 | 64 | - | - |
| $\psi$ | $2^{17}$ | 8 | 1024 | 2 | 64 | - | - | 2 | 64 |

Table 7: **Neural field architectures.** We provide the detailed parameter configurations of the neural fields we use to represent scene and object appearances.

function for the color head MLP$_{\mathbf{c}}$:

$$f(x) := \frac{1}{c}\text{sigmoid}(cx) \tag{11}$$

where $c := 0.9$ is a constant scaling factor. This allows the color prediction to slightly exceed the valid $[0, 1]$ RGB color space. After alpha compositing, we clamp the rendered RGB to the valid $[0, 1]$ range following [18].

**Time-dependent appearance.** In addition to conditioning the object appearance on the sequence at hand, we model the appearance of dynamic objects as a function of time by inputting $\gamma(t)$ to MLP$_{\mathbf{c}}$ as described above. This way, our method adapts to changes in scene lighting that are more intricate than the general scene appearance. This could be specular reflections, dynamic indicators such as brake lights, or shadows cast onto the object as it moves through the environment.

**Space contraction.** We use space contraction to query unbounded 3D Gaussian locations from the neural fields [54]. In particular, we use the following function for space contraction:

$$\zeta(\mathbf{x}) := \begin{cases} \mathbf{x}, & \|\mathbf{x}\| \le 1 \\ \left(2 - \frac{1}{\|\mathbf{x}\|}\right)\frac{\mathbf{x}}{\|\mathbf{x}\|}, & \|\mathbf{x}\| > 1 \end{cases}. \tag{12}$$

For $\phi$, we use $\|\cdot\|_\infty$ as the norm to contract the space, while for $\psi$ we use the Frobenius norm $\|\cdot\|_F$. Note that we use space contraction for $\psi$ because 3D Gaussians may extend beyond the 3D object dimensions to represent *e.g.* shadows, however, most of the representation capacity should be allocated to the object itself.

**Continuous-time object poses.** Both Argoverse 2 [81] and Waymo Open [23] provide precise timing information for both the LiDAR pointclouds to which the 3D bounding boxes are synchronized, and the camera images. Thus, we treat the dynamic object poses $\{\xi_o^{t_0}, ..., \xi_o^{t_n}\}$ as a continuous function of time $\xi_o(t)$, *i.e.* we interpolate between at $t_a \le t < t_b$ to time $t$ to compute $\xi_o(t)$. This also allows us to render videos at arbitrary frame rates with realistic, smooth object trajectories.

**Anti-aliased rendering.** Inspired by [59], we compensate for the screen space dilation introduced in [18] when evaluating $\mathfrak{g}_k^c$ multiplying by a compensation factor:

$$\mathfrak{g}_k^c(\mathbf{p}) := \sqrt{\frac{|\Sigma_k^c|}{|\Sigma_k^c + b\mathbf{I}|}} \exp\left(-\frac{1}{2}(\mathbf{p} - \boldsymbol{\mu}_k^c)^\top(\Sigma_k^c + b\mathbf{I})^{-1}(\mathbf{p} - \boldsymbol{\mu}_k^c)\right), \tag{13}$$

where $b$ is chosen to cover a single pixel in screen space. This helps us to render views at different sampling rates.

**Gradients of camera parameters.** Different from [18], we not only optimize the scene geometry but also the parameters of the camera poses. This greatly improves view quality in scenarios with imperfect camera calibration which is frequently the case in street scene datasets. In particular, we approximate the gradients w.r.t. a camera pose $[\mathbf{R}|\mathbf{t}]$ as:

$$\frac{\partial \mathcal{L}}{\partial \mathbf{t}} \approx -\sum_k \frac{\partial \mathcal{L}}{\partial \boldsymbol{\mu}_k}, \qquad \frac{\partial \mathcal{L}}{\partial \mathbf{R}} \approx -\left[\sum_k \frac{\partial \mathcal{L}}{\partial \boldsymbol{\mu}_k}(\boldsymbol{\mu}_k - \mathbf{t})^\top\right]\mathbf{R}. \tag{14}$$

This formulation was concurrently proposed in [61], so we refer to them for a detailed derivation. We obtain the gradients w.r.t. the vehicle poses $\xi$ via automatic differentiation [80].

**Adaptive density control.** We elaborate on the modifications described in Section 3.4. Specifically, we observe that the *same* 3D Gaussian will be rendered at varying but dominantly small scales. This biases the distribution of positional gradients towards views where the object is relatively small in

| Vanilla ADC | Modified ADC | Ground Truth |

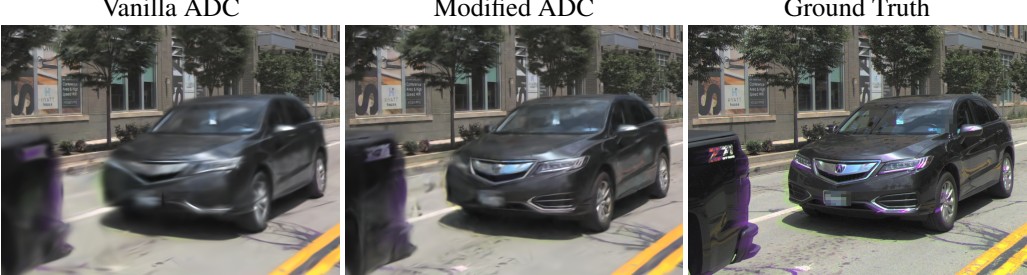

Figure 5: **Qualitative comparison of ADCs.** We show an example of a close-up car and observe over-smoothing when using vanilla ADC while our modified ADC leads to a sharper rendering result.

view space, leading to blurry renderings for close-ups due to insufficient densification. This motivates us to use maximum 2D screen size as an additional splitting criterion.

In addition to the adjustments described above and inspired by recent findings [60], we adapt the criterion of densification during ADC. In particular, Kerbl *et al.* [18] use the average absolute value of positional gradient $\frac{\partial \mathcal{L}}{\partial \boldsymbol{\mu}_k}$ across multiple iterations. The positional gradient of a projected 3D Gaussian is the sum of the positional gradients across the pixels it covers:

$$\frac{\partial \mathcal{L}}{\partial \boldsymbol{\mu}_k} = \sum_i \frac{\partial \mathcal{L}}{\partial \mathbf{p}_i} \frac{\partial \mathbf{p}_i}{\partial \boldsymbol{\mu}_k} \, . \tag{15}$$

However, this criterion is suboptimal when a 3D Gaussian spans more than a single pixel, a scenario that is particularly relevant for large-scale urban scenes. Specifically, since the positional gradient is composed of a sum of per-pixel gradients, these can point in different directions and thus cancel each other out. Therefore, we threshold

$$\sum_i \left\| \frac{\partial \mathcal{L}}{\partial \mathbf{p}_i} \frac{\partial \mathbf{p}_i}{\partial \boldsymbol{\mu}_k} \right\|_1 \tag{16}$$

as the criterion to drive densification decisions. This ensures that the overall magnitude of the gradients is considered, independent of the direction. However, this leads to an increased expected value, and therefore we increase the densification threshold to 0.0006.

**Hyperparameters.** We describe the hyperparameters used for our method, while training details can be found in Appendix A.3. For ADC, we use an opacity threshold of 0.005 to cull transparent 3D Gaussians. To maximize view quality, we do not cull 3D Gaussians after densification stops. We use a near clip plane at a 1.0m distance, scaled by the global scene scaling factor. We set this threshold to avoid numerical instability in the projection of 3D Gaussians. Indeed, the Jacobian $\mathsf{J}_k^c$ used in $\mathfrak{g}_k^c$ scales inversely with the depth of the primitive, which causes numerical instabilities as the depth of a 3D Gaussian approaches zero. For $\gamma(t)$, we use 6 frequencies to encode time $t$.

### A.3 Experiments

**Data preprocessing.** For each dataset, we obtain the initialization of the 3D Gaussians from a point cloud of the scene obtained from the provided LiDAR measurements. To avoid redundant points slowing down training, we voxelize this initial pointcloud with voxel sizes of $\tau := 0.1$m and $\tau := 0.15$m for the single- and multi-sequence experiments, respectively. We use the provided 3D bounding box annotations to filter points belonging to dynamic objects, to initialize the 3D Gaussians for each object, and as our object poses $\xi$.

For KITTI and VKITTI, we follow the established benchmark used in [16, 83, 17, 73]. We use the full resolution $375 \times 1242$ images for training and evaluation and evaluate at varying training set fractions. For Argoverse 2, we follow the experimental setup of [17]. In particular, we use the full resolution $1550 \times 2080$ images for training and evaluation and use all cameras of every 10th temporal frame as the testing split. Note that we used the provided masks from [17] to mask out parts of the ego-vehicle for both training and evaluation. For Waymo Open, we follow the experimental setup of EmerNeRF [76]. We use the three front cameras (`FRONT, FRONT_LEFT, FRONT_RIGHT`) and resize the images to $640 \times 960$ for both training and evaluation. We use only the first LiDAR return as initial points for our reconstruction. We follow [76] and evaluate the cameras of every 10th temporal frame.

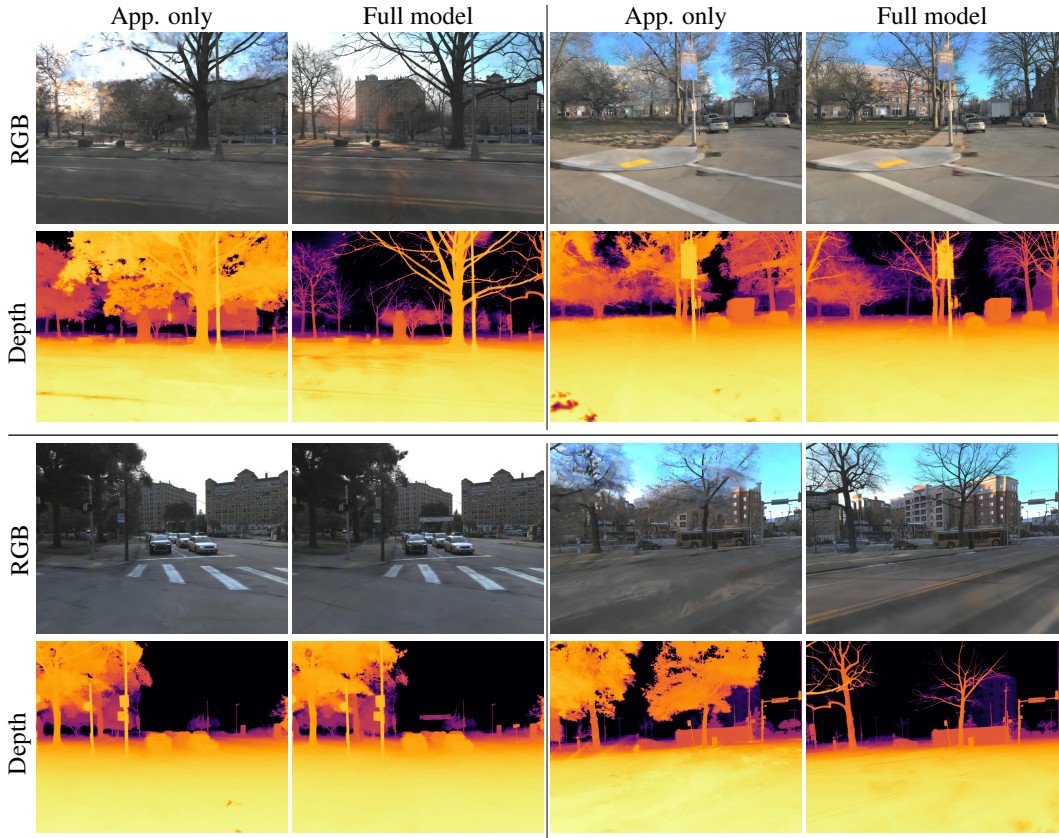

Figure 6: **Qualitative examples of transient geometry.** We show four relevant examples from the residential split of Argoverse 2 [81]. We observe a large disparity between our full model and ours without transient geometry modeling (App. only). Transient objects like a banner (left bottom) are completely missing and there are severe depth and color artifacts (*e.g.* trees). Best viewed digitally.

For separate evaluation of dynamic objects, we compute masks from the 2D ground truth camera bounding boxes. We keep only objects exceeding a velocity of 1 m/s to filter for potential sensor and annotation noise. We determine the velocities from the corresponding 3D bounding box annotations. Note also that [76] do not undistort the input images, and we follow this setup for a fair comparison.

**Implementation details.** For $\mathcal{L}_{\text{dep}}$, we use only the LiDAR measurements at the time of the camera sensor recording as ground truth to ensure dynamic objects receive valid depth supervision. We implement our method in PyTorch [80] with tools from nerfstudio [85]. For visualization of the depth, we use the `inferno_r` colormap and linear scaling in the 1-82.5 meters range.

During training, we use the Adam optimizer [86] with $\beta_1 \coloneqq 0.9, \beta_2 \coloneqq 0.999$. We use separate learning rates for each 3D Gaussian attribute, the neural fields, and the sequence latent codes $\omega_s^t$. In particular, for means $\boldsymbol{\mu}$, we use an exponential decay learning rate schedule from $1.6 \cdot 10^{-5}$ to $1.6 \cdot 10^{-6}$, for opacity $\alpha$, we use a learning rate of $5 \cdot 10^{-2}$, for scales $a$ and rotations $q$, we use a learning rate of $10^{-3}$. The neural fields are trained with an exponential decay learning rate schedule from $2.5 \cdot 10^{-3}$ to $2.5 \cdot 10^{-4}$. The sequence latent vectors $\omega_s^t$ are optimized with a learning rate of $5 \cdot 10^{-4}$. We optimize camera and object pose parameters with an exponential decay learning rate schedule from $10^{-5}$ to $10^{-6}$. To counter pose drift, we apply weight decay with a factor $10^{-2}$. Note that we also optimize the height of object poses $\xi$. We follow previous works [87, 85, 17] and optimize the evaluation camera poses when optimizing training poses to compensate for pose errors introduced by drifting geometry through optimized training poses that may contaminate the view synthesis quality measurement.

In our multi-sequence experiments in Table 1 and Table 5, we train our model on 8 NVIDIA A100 40GB GPUs for 125,000 steps, taking approximately 2.5 days. In our single-sequence experiments, we train our model on a single RTX 4090 GPU for several hours. On Waymo Open, we train our model for 60,000 steps while for KITTI and VKITTI2 we train the model for 30,000 steps. For

| Split | 3D Box Type | PSNR ↑ | SSIM ↑ | LPIPS ↓ |
|---|---|---|---|---|
| Single Seq. | GT | **28.81** | **0.845** | **0.260** |
| | Prediction | 28.52 | 0.842 | 0.264 |
| Multi. Seq. | GT | **25.78** | **0.772** | **0.405** |
| | Prediction | 25.67 | **0.772** | 0.409 |

(a) **Ablation on 3D bounding boxes**. We use 3D box annotations and predictions from an off-the-shelf 3D tracker.

| | Full Image | | Dynamic-Only | |
|---|---|---|---|---|
| $MLP_\chi$ | PSNR ↑ | SSIM ↑ | PSNR ↑ | SSIM ↑ |
| - | 29.52 | 0.891 | 30.08 | 0.895 |
| ✓ | **29.60** | **0.892** | **30.15** | **0.896** |

(b) **Ablation on deformation head** $\chi$. We compare results with and without modeling non-rigid motion as deformations.

Table 8: **Addtional ablation studies.** In (a) we show results on a single sequence of the residential area in our benchmark and the full residential area. In (b) we use a subset of the Dynamic-32 split from [76], *i.e.* the 12 sequences with the highest number of non-rigid objects.

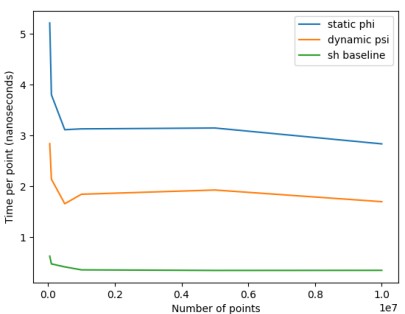

Figure 7: **Runtime comparison of neural fields vs. spherical harmonics.** We compare the runtime of querying neural fields $\phi$ and $\psi$ to a spherical harmonics function of degree 3. We report time-per-query in nanoseconds.

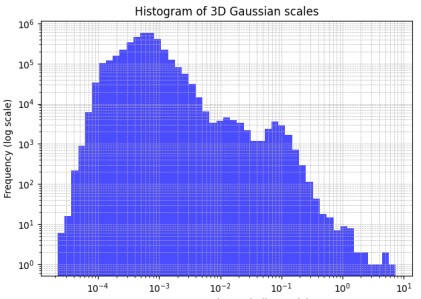

Figure 8: **Histogram of mean 3D Gaussian scales.** We use our model trained on Argoverse 2 (residential split). Both axes are in *logarithmic* scale. The vast majority of 3D Gaussians have a small scale, while there are a few outliers with huge scales. The scene is approximately within [-1, 1].

our single-sequence experiments in Table 5 we use a schedule of 100,000 steps. We chose a longer schedule for Waymo Open and Argoverse 2 since the scenes are more complex and contain about $5 - 10\times$ as many images as the sequences in KITTI and VKITTI2.

We linearly scale the number warm-up steps, the steps per ADC, and the maximum step to invoke ADC with the number of training steps. For multi-GPU training, we reduce these parameters linearly with the number of GPUs. However, we observed that scaling the learning rates linearly does perform subpar to the initial learning rates in the multi-GPU setup, and therefore we keep the learning rates the same across all experiments.

**Additional ablation studies.** In Table 8a, we show that while our approach benefits from high-quality 3D bounding boxes, it is robust to noise and achieves a high view synthesis quality even with noisy predictions acquired from a 3D tracking algorithm [88]. In Table 8b, we demonstrate that the deformation head yields a small, albeit noticeable improvement in quantitative rendering results. This corroborates the utility of deformation head $\chi$ beyond the qualitative examples shown in Figures 4 and 10. Note that the threshold to distinguish between dynamic and static areas is 1m/s following [76] so that some instances like slow-moving pedestrians will be classified as static. Also, since non-rigid entities usually cover only a small portion of the scene, expected improvements are inherently small.

In Table 9, we show that our modified ADC increases view quality in general, and *perceptual* quality in particular as it avoids blurry close-up renderings. Note that our ADC leads to roughly twice the number of 3D Gaussians belonging to objects compared

| Mod. ADC | PSNR ↑ | SSIM ↑ | LPIPS ↓ | Num. Total | Num. Obj. |
|---|---|---|---|---|---|
| - | 28.60 | 0.834 | 0.270 | 3.72M | 314K |
| ✓ | **28.81** | **0.845** | **0.260** | 4.1M | 611K |

Table 9: **Ablation study on ADC.** We compare the performance of vanilla ADC to our modified variant in the single-sequence setting.

to vanilla ADC, thus avoiding insufficient densification. We also show a qualitative example in Figure 5, illustrating this effect. The close-up car rendering is significantly sharper using the modified ADC. Note that for both variants, we use the absolute gradient criterion (see Appendix A.2) for a fair comparison.

**Additional runtime and model analysis.** In Figure 7, we provide a comparison in terms of time per query of neural fields of different sizes (equivalent to the sizes of our $\phi$ and $\psi$) versus querying a spherical harmonics function of degree 3 as used in 3DGS [18]. We observe that the SH function is approx. 6 to $8\times$ faster to evaluate than the neural fields. However, we show that this does not lead to a critical increase in overall runtime in Table 6. Additionally, we note that the SH function is limited in representation capacity: It is not capable of handling varying appearance across input sequences (weather, time of day, season), transient geometry (construction sites, tree leaves), articulated motion of dynamic objects (pedestrians, cyclists), and large-scale scenes with several millions of 3D Gaussians due to memory constraints. Thus, we emphasize that the use of neural fields does not merely improve memory footprint, but enables applications that 3DGS [18] is not capable of modeling.

In Figure 8, provide an analysis on the distribution of 3D Gaussian scales across a large-scale urban scene reconstruction. We notice that while the vast majority of 3D Gaussians is small ($< 0.001$ mean scale), there is a small number of very large 3D Gaussians ($> 1.0$ mean scale) compared to the scene bounds that are approximately within [-1.0, 1.0]. This can lead to fog-like artifacts in free-viewpoint renderings. We note that this can be mitigated using a regularization term on the 3D Gaussian scales, however, we did not observe an quantitative improvement in view synthesis metrics and thus we did not include it in the experiments.

**Qualitative results.** We provide an additional qualitative comparison of the variants iii) and ii) introduced in Section 4.2, *i.e.* our model with and without transient geometry modeling. In Figure 6, we show multiple examples confirming that iii) indeed models transient geometry such as tree leaves or temporary advertisement banners (bottom left), and effectively mitigates the severe artifacts present in the RGB renderings of ii). Furthermore, the depth maps show that iii) faithfully represents the true geometry, while ii) lacks geometric variability across sequences.

In addition, we show qualitative comparisons to the state-of-the-art in Figure 9. Our method continues to produce sharper renderings than the previous best-performing method [17], while also handling articulated objects such as pedestrians which are missing in the reconstruction of previous works (bottom two rows). Finally, we show another temporal sequence of evaluation frames in Figure 10. Our method handles unconstrained motions and can also reconstruct more complicated scenarios such as a pedestrian carrying a stroller (right), or a grocery bag (left).

|  SUDS [16]  |  ML-NSG [17]  |  4DGF (Ours)  |  Ground Truth  |

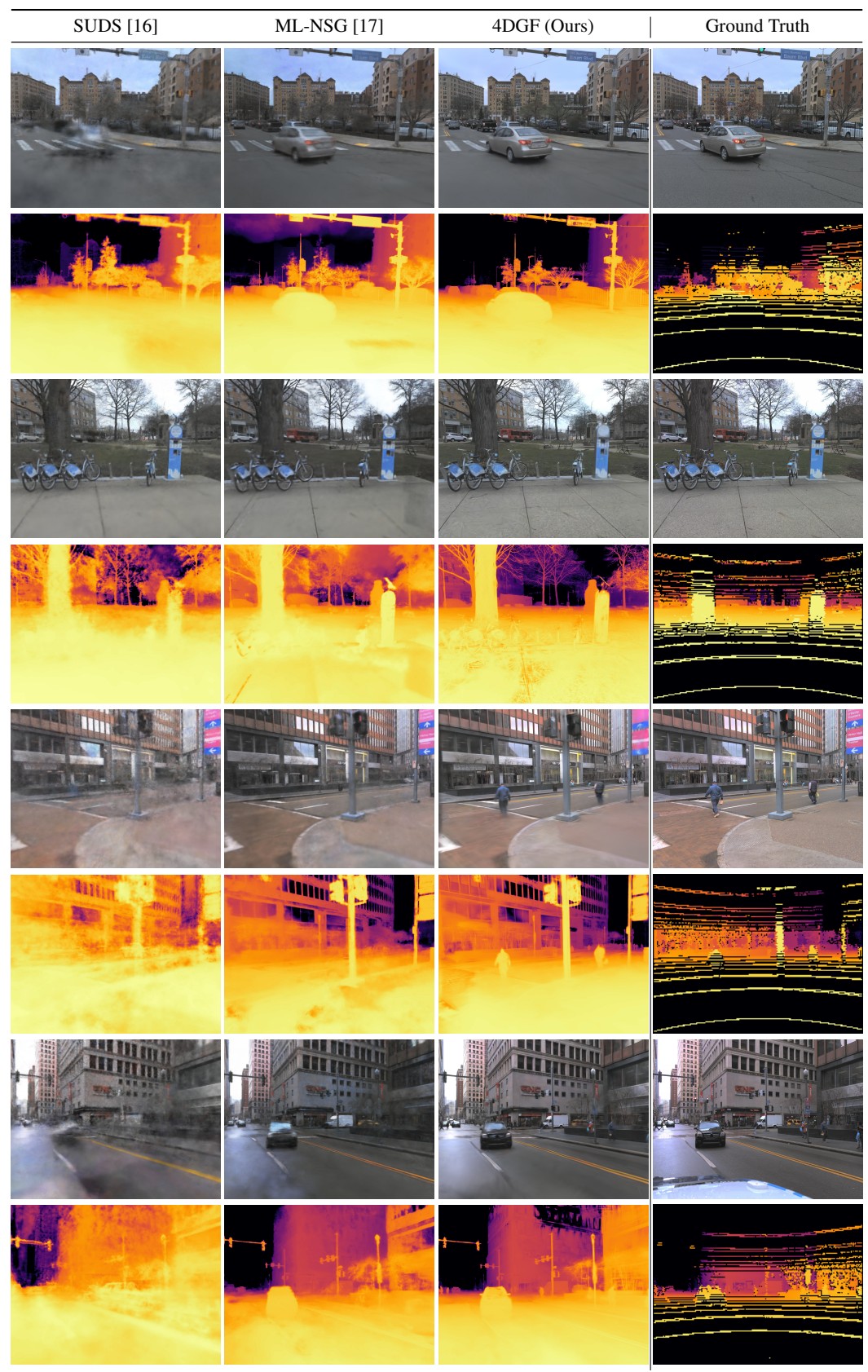

Figure 9: **Additional qualitative results on Argoverse 2 [81].** We show four examples where the upper two are from the residential area and the lower two are from the downtown area.

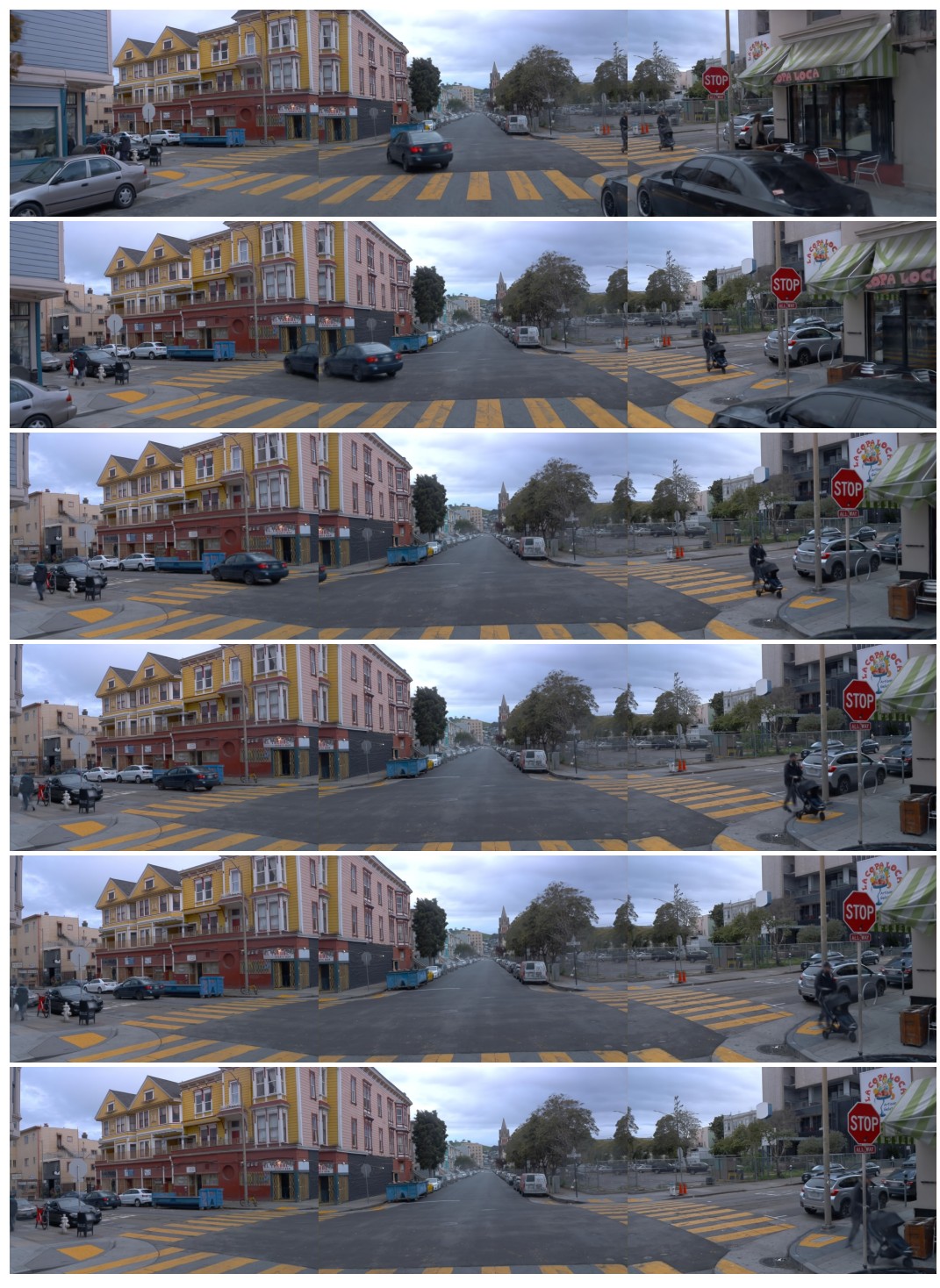

Figure 10: **Additional qualitative results on Waymo Open [23].** We show a sequence of evaluation views synthesized by our model (top to bottom). We see two pedestrians on the left and right being faithfully modeled across varying body poses while also carrying objects such as a stroller (right) or a shopping bag (left).

