# OpenReview forum: "Dynamic 3D Gaussian Fields for Urban Areas"
_NeurIPS.cc/2024/Conference — NeurIPS 2024 spotlight_

### Official Review · Reviewer_51SN · 2024-07-11

**Soundness:** 3
**Presentation:** 2
**Contribution:** 2
**Rating:** 5
**Confidence:** 5

**Summary:**

This paper aims to perform view synthesis for dynamic urban scenes. This paper adopts 3DGS as scene geometry and uses neural fields to model the dynamic appearance of urban scenes. The neural scene graph is introduced to handle the movement of dynamic objects, and a deformation field is used to handle local articulated motions. Experiments show that the proposed approach outperforms baseline methods.

**Strengths:**

1. The presented pipeline well handles the dynamic appearance of urban scenes.
2. The experiments are sufficient and validate the effectiveness of the proposed approach.
3. The idea of combining neural fields with 3DGS is sound and effective.

**Weaknesses:**

1. The method presented in the paper takes 0.17 seconds to render an image at a resolution of 1550x2048, which is significantly slower than conventional 3DGS. Is the trade-off of such a significant sacrifice in rendering speed for quality improvement justified? Does the author have any solutions to address this issue?
2. The paper needs to evaluate the extent to which neural fields impact the rendering speed of 3DGS.
3. The pipeline figure of the paper should be clearer. The connections between the various modules are not easily discernible from the figure and its caption. For instance, it is not clearly depicted how the latent codes obtained from the scene configuration are inputted into the neural fields. Then, how are neural fields combined with 3DGS to represent static scenes and dynamic objects? The figure only shows simple association arrows. However, these modules are not merely input-output relationships. There are some combination operations between them.
4. The paper uses neural fields to represent appearance, which reduces the memory footprint but may also significantly impact rendering speed. Has the paper considered how to address this issue?
5. In Figure 2 of the paper, regarding the neural fields section, the symbols for static opacity correction and dynamic deformation are inconsistent with the descriptions in Section 3.2 of the paper. This is quite confusing.
6. I am curious whether the combination of neural fields with 3DGS could make the optimization of 3DGS unstable?
7. The non-rigid objects mentioned in the paper refer to cars, right? Or other objects? I did not see how the paper describes the modeling of cars. Although the paper mentions the use of scene graphs for modeling, I did not see how dynamic cars are represented using scene graphs. Does the paper treat dynamic cars as non-rigid objects directly? In this case, how can the large range of movement of dynamic cars be handled?

**Questions:**

The presentation of this paper should be improved. Some important technical details are missing. The limitations from the introduction of neural fields should be discussed.

**Limitations:**

The limitations from the introduction of neural fields should be discussed.

---

> ### Author Rebuttal · Authors · 2024-08-07
>
> We thank the reviewer for the helpful feedback and for taking the time to review our manuscript. Below we address the concerns raised.
> 1. Rendering speed compared to 3DGS: We hope to address this concern in our global response, where we show a) an improved runtime of 0.074 seconds when rendering frames at 1550x2048 resolution using 8.02M 3D Gaussians, and b) a detailed runtime analysis of our method. This analysis reveals that the rasterization and the scene graph evaluation necessary for rendering dynamic scenes account for more than 75% of the total runtime. While we acknowledge that rendering large-scale urban dynamic scenes is more complex than rendering small-scale static 3D scenes and thus the reported rendering speeds are significantly slower than in 3DGS [18], we believe that the complex nature of the scenes we render needs to be taken into consideration when examining these numbers.
> 2. Figure 2 unclear: We plan to improve the clarity of the figure for the camera-ready version, highlighting the information flow. To render an image at sequence $s$ and time $t$, we first evaluate the scene graph which returns the latent codes (i.e. the conditioning signals for the neural fields), as well as coordinate transformations and the active set of 3D Gaussians (i.e. which objects are present where). We then use the latent codes in conjunction with the sets of 3D Gaussians as input to the neural fields to retrieve the 3D Gaussian color and the opacity correction or deformation. After this, we use the coordinate transformations to compose the scene globally, i.e. placing the objects at the right locations. Finally, we render the scene with the 3DGS renderer.
> 3. Impact of neural fields on rendering speed: In our global response above, we show that querying the neural fields accounts for only 13% of the total rendering speed. This is because we use efficient hash-grid based neural fields inspired by [53]. Therefore, although the neural fields are slower to query than the spherical harmonics used in 3DGS [18], it does not critically impact rendering speed.
> 4. Figure 2 symbols: We thank the reviewer for pointing this out, indeed we abbreviated $\delta \mu_k^t$ in the figure to $\delta$ to make it more concise. We will correct this in the camera-ready version.
> 5. Impact of neural fields on optimization: In our experiments, the optimization was not negatively impacted by the introduction of neural fields. In particular, we show in Tab. 4a that replacing the SH color function with neural fields does not lead to a significant change in performance while halving the GPU memory requirements (row 2 vs. 3).
> 6. Dynamic vs. non-rigid objects: We apologize for the confusion. In L196-197 of the paper, we refer to vehicles (e.g. cars) as rigid dynamic objects since their motion can be described with a single rigid body transformation, while we refer to pedestrians, cyclists, and others as non-rigid dynamic objects since their motion is articulated, i.e. cannot be represented by a single rigid body transformation. We represent the global motion of all dynamic objects with our scene graph, transforming the objects from an object-centric canonical space to world space with rigid body transformations. Additionally, we represent the local, articulated motion of non-rigid objects like pedestrians as deformation in canonical space. We hope this clarifies the reviewer’s concern and we will adjust our writing in the camera-ready version accordingly.
>
> We hope that our response adequately addresses the reviewer’s concerns. We are happy to provide more information during the discussion phase.

---

> > ### Comment · Reviewer_51SN · 2024-08-09
> >
> > After reading the rebuttal and other reviewers' comments, I think that this paper reaches the bar of NeurIPS.

---

### Official Review · Reviewer_7pd2 · 2024-07-12

**Soundness:** 3
**Presentation:** 4
**Contribution:** 3
**Rating:** 6
**Confidence:** 5

**Summary:**

This paper proposes a hybrid neural scene representation for dynamic urban driving scene modelling. The method utilizes 3D Gaussians as an efficient geometric scaffold and neural fields to represent appearance, thereby reducing memory. To account for transient scene geometry variations caused by weather, seasons, and other factors, the authors introduce an opacity attenuation field that modulates the scene geometry. For modeling dynamic actors in the scene, an object-centric representation is used, with a non-rigid deformation in the canonical space to animate objects such as pedestrians. Experiments demonstrate that the proposed method achieves state-of-the-art performance while rendering faster than previous methods.

**Strengths:**

* The paper is well-written and easy to follow.
* The decomposed representation of appearance significantly reduces memory usage.
* It models transient scene appearance and geometry, as well as non-rigid objects like pedestrians.
* The evaluation and ablation study are comprehensive.
* The paper demonstrates visually superior results compared to baselines such as SUDS and ML-NSG.

**Weaknesses:**

* The rendering of the proposed scene representation requires query appearance from the neural fields, it is unclear whether this will impact rendering speed compared to spherical harmonics representation.
* This paper lacks a comparison with recent neural field baselines such as UniSim and NeuRAD for urban driving scenes. Additionally, there is no comparison of the speed to 3D Gaussian baselines.
* How to control the non-rigid objects in the scene? e.g., animating the pedestrians given a sequence of human poses.
* Is it feasible to render other sensor modalities in autonomous driving, such as LiDAR?

**Questions:**

This paper addresses a practical and important problem in autonomous driving. The writing is clear, and the results are promising. I look forward to the authors' response to the concerns I raised above.

**Limitations:**

The authors discussed the limitations of modeling other sensor phenomena such as rolling shutter effects, motion, and more complex camera models. They also discussed the broader societal impacts of their work.

---

> ### Author Rebuttal · Authors · 2024-08-07
>
> We thank the reviewer for taking the time to review our manuscript and the constructive feedback on our work. Below we address the concerns raised.
> 1. Impact of neural field query on rendering speed: We address this point in the global rebuttal posted above. In a nutshell, we show that the neural field query accounts for only 13% of the total runtime, so while the neural field query is slower compared to spherical harmonics, the impact on the rendering speed is marginal.
> 2. Additional comparisons: We thank the reviewer for pointing us to UniSim and NeuRAD. While at submission time, we were not aware that the code for either of the two papers was public, we noticed that the code for NeuRAD was released by now. Therefore, we provide an additional comparison on KITTI MOT in Tab. 2 of the PDF attached to the global response and show that our method outperforms NeuRAD. Regarding further comparison to 3D Gaussian baselines, we refer the reviewer to Tables 2 and 4a of our paper, where we compare to StreetGaussians [73] and the SH color representation used in 3DGS [18] (row 2 vs. 3), respectively.
> 3. Control of non-rigid objects: We build a model capable of coping with general dynamic urban scenarios, which exhibit a wide range of dynamic actors. Thus, the design of our method is such that it allows for general modeling of non-rigid objects, e.g. pedestrians holding a stroller or shopping bags (see Fig. 8 in our supplemental material for an example), cyclists, and animals. Specifically, the articulated motion of pedestrians is controlled via time input to the deformation head, not via human pose estimates.
> 4. LiDAR rendering: While we did not consider rendering LiDAR measurements in our work, it is possible with our representation. However, we note that the 3DGS renderer assumes a pinhole camera model, which differs from the LiDAR sensor model. Therefore, ray-tracing-based approaches for rendering 3D Gaussians should be utilized [Yu et al.]. In addition, the LiDAR intensity and ray drop probability should be modeled with the neural fields following e.g. NeuRAD. The LiDAR point cloud can then be rendered from rays generated by the LiDAR sensor model. We consider this as an interesting direction to explore for future work.
>
> We hope this adequately addresses the reviewer’s concerns. We are happy to provide more information during the discussion phase.
>
> [Yu et al.] Gaussian Opacity Fields: Efficient and Compact Surface Reconstruction in Unbounded Scenes

---

### Official Review · Reviewer_6enk · 2024-07-13

**Soundness:** 4
**Presentation:** 4
**Contribution:** 3
**Rating:** 7
**Confidence:** 5

**Summary:**

The paper presents a novel 3D scene representation for novel view synthesis (nvs) in dynamic urban environments where, in particular, under heterogeneous imaging environments. The proposed representation relies on existing ingredients: 3D Gaussian Splatting, learned static/dynamic object instances, and  a global scene graph.

The resulting system yields very strong results on a series of public autonomous driving benchmarks.

**Strengths:**

### + Readability.
Overall, in its current state, the paper's readability is relatively good. The main ideas, concepts, are mostly well discussed, conveyed, and articulated, throughout the paper.

### + Practical usefullness of the considered problem.

### + Structure, and Organization of the Contents.
The presentation is mostly on point and each dedicated section of the paper is properly balanced. The use of text real-estate is fair.

### + Relative simplicity of the conceptual contribution.

### + The amount of implementation details is very good.

### + The reported performance.

### + Implementation details for reproducibility: excellent.

**Weaknesses:**

### - (1) Positioning of the conceptual contribution vs. the competitive landscape.

In particular, the proposed method looks very much like a revisit of Panoptic Radiance Fields [49] by replacing the NeRF component byt 3D Gaussian splats.

While this is perfectly fine, this merits a targeted, transparent discussion in the main paper to help the reader understand the whereabouts of how the proposed contribution relates (or not) with such pieces of litterature.

### - (2) How much does it cost?

Missing piece of information regarding the resource usage, memory footprint, typical timings etc to better understand the downsides of using the provided method.

### - (3) (To a lesser extent) Certain contents in the paper are unclear.

Figure 4: what is happening? Adding color annotations or boxes would definitely help.

**Questions:**

I do not have more questions or suggestions than the ones underlying the aforementioned weaknesses.

**Limitations:**

The authors provide one dedicated paragraph that reasonably addresses such considerations.

---

> ### Author Rebuttal · Authors · 2024-08-07
>
> We thank the reviewer for the helpful feedback and for taking the time to review our manuscript. Below we address the concerns raised.
> 1. Positioning of the conceptual contribution vs. the competitive landscape: We addressed this in  L114-117 of our paper, but we will make the distinction clearer in the camera-ready version. Specifically, while our work shares similarities to [49] in how the rigid, global motion of dynamic objects is handled, our method fundamentally differs in several key aspects: 1. We introduce a mechanism to handle both transient geometry and varying appearance across multiple, *heterogeneous* captures, a key component that [49] is not addressing 2. This mechanism and the efficiency of our approach enable the reconstruction of much larger urban areas with significantly more dynamic objects than in [49], 3. While [49] excludes non-rigid objects, we model the articulated motion of non-rigid dynamic objects, and 4. We do not rely on semantic priors to represent dynamic objects accurately.
> 2. Runtime/memory cost: We address it in our global response. In short, we show that the neural fields introduced by our method do not dominate the runtime. We report peak memory consumption in Tab. 4a and show that our method leads to a reduced memory footprint compared to the SH color representation in 3DGS [18].
> 3. Figure 4: Thank you for pointing out this issue. The important bit is happening in the middle of the image: The lady in the pink sweater is getting out of the dark grey car in front of the ego-vehicle. It illustrates the ability of our method to handle complex, articulated motions. We will add colored boxes to the camera-ready version to make it clear.
>
> We hope our response adequately addresses the reviewer’s concerns. We are happy to provide more information during the discussion phase.

---

> > ### Comment · Reviewer_6enk · 2024-08-13
> > **Response to Rebuttal**
> >
> > Dear Authors,
> >
> > Having read (all of) the rebuttal contents, here is my response.
> >
> > Most of my concerns have been adequately addressed but there is one misunderstanding regarding point 2 in your reply:
> > My point was refering to runtime and resource usage numbers wrt the competition, in particular [49] with indicative metrics, even at the top level.
> >
> > 1 and 2 above address this without typical ranges, this is what is missing and I would strongly suggest to integrate one line in the camera ready version to clarify this as it is an additional plus favoring the proposed contribution.
> >
> > I will maintain my positive rating.
> >
> > Warm regards.

---

> > > ### Author Response · Authors · 2024-08-14
> > >
> > > We thank the reviewer for the clarification. We will integrate the line clarifying the runtime and resource usage compared to the competition including typical ranges in the camera ready version as suggested.

---

### Official Review · Reviewer_f8M6 · 2024-07-15

**Soundness:** 3
**Presentation:** 4
**Contribution:** 3
**Rating:** 8
**Confidence:** 4

**Summary:**

This paper works on novel view synthesis (NVS) for large-scale, dynamic urban scenes. This paper proposes a neural scene representation called 4DGF, which uses 3D Gaussians as an efficient geometry scaffold while relying on neural fields as a compact and flexible appearance model. The proposed method integrates scene dynamics via a scene graph at global scale while modeling articulated motions on a local level via deformations. The method significantly outperforms baselines in terms of speed and rendering quality on three benchmarks.

**Strengths:**

1. The idea of combining Gaussian Splatting and neural fields to model geometry and appearance, respectively, is very interesting. This makes a lot of sense considering the efficiency and the advantages of each of the two representation. This is definitely a more scalable approach to large-scale scenes compared to prior work.

2. Extensive experiments have been conducted to validate the proposed method, this includes comparing with recent baselines on three benchmarks and the ablation studies that carefully examine each component. Moreover, the rendering quality improvement and the speedup is very significant on all three datasets.

3. The paper is very well-written and easy to follow. Implementation details are sufficiently discussed for reproducibility.

**Weaknesses:**

1. I appreciate the authors' including a video in the submission. I found sometimes there's a large foggy region near the camera (e.g., the regions on the right during the 5-6th second), do the authors have any explanations on that? Is it caused by any limitations discussed in Sec. 5?

2. I understand that this paper mainly focuses on large dynamic scenes. I am curious how this hybrid representation performs on 3D statics scenes (e.g., the benchmarks that the original 3DGS have been tested on). This seems to be a more straightforward way to see the effect of using neural fields instead to model appearance.

**Questions:**

Minor questions/suggestions:

What does GPU memory in Tab. 4 (a) mean? Is it peak memory?

In Fig. 1 inputs, the blue/orange colors for the image boundaries are also used for "geometry" and "appearance" respectively. I assume there's not such a correspondence between the input and geometry/appearance. So maybe you could change the input boundaries to different colors.

FIg. 2: extra space in "Scene configuration"

Tab. 4 appears before Tab. 3, maybe you could switch the order.

**Limitations:**

The limitations seem to have been sufficiently discussed.

---

> ### Author Rebuttal · Authors · 2024-08-07
>
> We thank the reviewer for taking the time to review our manuscript and the constructive feedback on our work. Below we address the concerns raised.
> 1. Foggy regions in the demonstration video: While these artifacts may be caused by the limitations discussed in Sec. 5 like white balance or focus blur, we suspect it is due to an issue with 3DGS optimization. In particular, we noticed that 3DGS tends to produce a small number of very large, semi-transparent 3D Gaussians, especially when pruning thresholds are set generously. In Fig. 1 of the PDF attached to the global response, we show there are a few 3D Gaussians with a huge mean scale (>1.0), while the vast majority is small (<0.001 mean scale) compared to the scene bounds that are approximately within [-1.0, 1.0]. Concurrently proposed techniques like scale regularization [Kheradmand et al.] could mitigate this issue.
> 2. Experiments on static 3D scenes: In Tab. 4a, we show that our method performs similarly when using neural fields as color representation compared to the SH representation from 3DGS [18] (row 2 vs. 3). This illustrates that, on single-sequence, homogeneous data, SH and neural fields are equivalently suitable for modeling appearance, both for static and dynamic scene parts. This finding is in line with concurrent work focusing on static 3D scenes [Lu et al.]. In contrast, Tab. 4b shows that, on multi-sequence, heterogeneous data, modeling appearance and transient geometry with neural fields is essential for NVS performance. However, we agree this is an interesting experiment and will add this comparison to the camera-ready version.
> 3. GPU memory in Tab. 4a: We report peak memory consumption as it is the deciding factor when attempting to train a 3DGS model on a specific scene.
> 4. Suggestions on Fig. 1/2, Tab. 3/4: Thank you for spotting these issues. We will correct them in the camera-ready version.
>
> We hope our response adequately addresses the reviewer’s concerns. We are happy to provide more information during the discussion phase.
>
> [Kheradmand et al.] 3D Gaussian Splatting as Markov Chain Monte Carlo
>
> [Lu et al.] Scaffold-GS: Structured 3D Gaussians for View-Adaptive Rendering

---

> > ### Comment · Reviewer_f8M6 · 2024-08-13
> >
> > I have read through all reviewers' comments and the rebuttals by authors. My concerns have been addressed, and I would like to keep my original score.

---

### Author Rebuttal · Authors · 2024-08-07

We thank all reviewers for their helpful and constructive feedback. We appreciate the positive reception of our work. The consensus is that the combination of neural fields and 3D Gaussians as scene representation constitutes an interesting (f8M6), conceptually simple (6enk), and effective (51SN) solution to practical problems with 3DGS [18], i.e. memory consumption and thus scalability to large-scale scenes (f8M6, 7pd2) as well as modeling of complex real-world phenomena such as transient geometry and appearance across captures and non-rigid objects (7pd2). The reviewers point out that the improvements over existing work are convincing with visually superior results compared to prior art (f8M6, 7pd2, 6enk), and acknowledge the comprehensive evaluation and ablation studies (f8M6, 7pd2, 51SN) validating the effectiveness of our approach. Finally, most reviewers agree that the paper is well-written, easy to follow (f8M6, 6enk, 7pd2), and the presentation is mostly on point (6enk). The reviewers also acknowledge the extensive discussion of experimental and implementation details for reproducibility (f8M6, 6enk).

A shared concern among the reviews is that a runtime analysis should be included. We are thankful for this suggestion and a) include a detailed runtime analysis in the attached PDF, and b) *improve the runtime of our algorithm by more than 2x*. In particular, we implement two improvements:  First, we significantly reduce the number of queries to the neural fields by skipping out-of-view 3D Gaussians after projection. Second, we implement a vectorized version of the neural field query that retrieves the outputs for all dynamic objects in parallel. Next, we measure the runtime of each pipeline component and report the results in Tab. 1. It shows that the runtime is dominated by rasterization and scene graph evaluation, accounting for more than 75% of the total average runtime, not by the neural field queries (approx. 13%). This is because we render views at high resolution with a large number of 3D Gaussians which is computationally demanding even for the efficient rasterization of 3DGS [18], and because we represent areas with hundreds to thousands of dynamic objects, making the retrieval of the 3D Gaussians and latent codes costly. For the neural fields, we use very efficient hash-grid based encodings in conjunction with tiny MLPs, inspired by InstantNGP [53]. Thus, simply replacing the neural fields with the SH function of [18] would not lead to a significant speedup. Note also that our method is significantly faster in simpler scenarios with fewer 3D Gaussians per scene, e.g. Waymo Open.
In addition, we provide a comparison in terms of time per query of the SH function used in 3DGS [18] versus our hash-grid based neural field query. While the SH function is faster to evaluate than the neural fields, it is also severely limited in representation capacity: It is not capable of handling varying appearance across input sequences (weather, time of day, season),  transient geometry (construction sites, tree leaves), articulated motion of dynamic objects (pedestrians, cyclists), and large-scale scenes with several millions of 3D Gaussians due to memory constraints. Thus, we emphasize that the use of neural fields does not merely improve results, but *enables applications that 3DGS [18] is not capable of modeling*. We will dedicate a paragraph to the runtime analysis in the camera-ready version.

---

### Author Response · Authors · 2024-08-14

Dear Reviewers,

We are grateful for the several suggestions and pointers to improve our paper and are happy that we could address the concerns raised. Thank you once again for reviewing our manuscript and participating in the discussion.

Best regards,
The Author(s)

---

### Decision · Program_Chairs · 2024-09-25

**Decision:**

Accept (spotlight)

**Comment:**

The paper proposed a new 3D scene representation that combines the strength of 3DGS and neural field to model dynamic urban scenes, which leads to better efficiency, lower memory consumption, and higher quality on benchmark datasets, while being able to handle the transient geometry and appearance. The decomposition of dynamic and static components further helps to model non-rigid motion, such as pedestrians. These are valuable contributions to our research community.

The paper received mixed reviews initially, but the rebuttal and following discussions successfully addressed the concerns of all the reviewers who all became positive after the rebuttal. After discussing with SCA, the AC agrees with the reviewers' opinions on accepting this paper.  As promised during the rebuttal and discussion, please add the additional results of running time comparisons and analysis provided during the rebuttal to the main paper.